# TOWARDS REASONABLE CONCEPT BOTTLENECKS

## ABSTRACT

We propose a novel, flexible, and efficient framework for designing Concept Bottleneck Models (CBMs) that enables practitioners to explicitly encode any of their prior knowledge and beliefs about the concept-concept (`C-C`) and concept-task (`C→Y`) relationships into the model reasoning. The resulting **C**oncept **REA**soning **M**odels (CREAMs) architecturally encode potentially sparse `C→Y` relationships, as well as various types of `C-C` relationships such as mutual exclusivity, hierarchical associations, and/or correlations. Moreover, CREAM can include a regularized side-channel to complement the potentially incomplete concept sets, achieving competitive task performance while encouraging predictions to be concept-grounded. Our experiments show that, without additional computational overhead, the CREAM designs: (i) allow for efficient and accurate interventions by avoiding leakage; and (ii) achieve task performance on par with black-box models.

## 1 INTRODUCTION

Deep neural networks (DNNs) have become ubiquitous in various aspects of our daily lives but their opaque decision-making limits transparency, user understanding, and trust. Interpretability is essential for reliable AI, especially in finance (Doshi-Velez & Kim, 2017), healthcare (Rudin, 2019), autonomous systems (Samek et al., 2019) (Doshi-Velez & Kim, 2017; Lipton, 2018), and so forth.

Interpretable models have therefore gained attention (Molnar, 2025; Rudin et al., 2022), particularly concept-based approaches that explain predictions through human-understandable concepts (Barbiero et al., 2023; Chen et al., 2020; Koh et al., 2020; Oikarinen et al., 2023; Poeta et al., 2023; Yeh et al., 2020; Yuksekgonul et al., 2023). Concept Bottleneck Models (CBMs) (Koh et al., 2020) exemplify this by introducing an intermediate concept layer, where concepts are explicitly learned and predicted prior to the task, enabling *transparent reasoning* and *human intervention*. CBMs have been applied on various fields, including medical diagnosis (Daneshjou et al., 2022), predictive maintenance (Forest et al., 2024), and vision-language tasks (Yang et al., 2023).

Standard CBMs assume conditional independence among concepts, limiting their ability to model intra-concept or concept–task relationships (Dominici et al., 2025), a property we call *structured model reasoning*. They also assume the concept set is complete and sufficient for prediction. Extensions have relaxed these assumptions (Dominici et al., 2025), but typically in problem-specific ways that trade one limitation for another. Real-world datasets often exhibit *concept incompleteness*, which reduces accuracy (Grivas et al., 2024; Yeh et al., 2020). Moreover, even with correct concept predictions, models may exploit unintended information, called *concept leakage*, to bypass intended reasoning pathways (Mahinpei et al., 2021; Margeloiu et al., 2021), undermining interpretability and encouraging misplaced trust (Marconato et al., 2023a).

We propose **C**oncept **REA**soning **M**odels (CREAM), a framework for CBMs that encodes prior knowledge of `C-C` and `C→Y` relations through a reasoning graph. The graph combines hard constraints (e.g., blocking irrelevant edges, enforcing exclusivity) with probabilistic dependencies learned from data. By embedding these relations into CREAM, predictions are grounded in a user-specified reasoning graph. Each concept influences only a sparse subset of predictions and any given output can be traced back to a limited set of candidate concepts, ensuring tractability and enhancing interpretability and intervenability while mitigating leakage. A regularized side-channel captures supplementary task-relevant information without compromising concept-based predictions, unlike prior work (Dominici et al., 2025). Importantly, the `C-C`, `C→Y` blocks, and side-channel are independent modular components that can each be modified, included, or excluded separately, depending on the

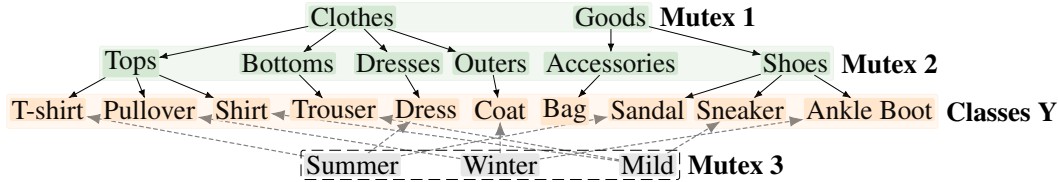

Figure 1: Reasoning graph for FMNIST from (Seo & Shin, 2019). We show the incomplete concept set used in *iFMNIST* as well as the additional season-related concepts from *cFMNIST*. Concepts and classes within the boxes are mutually exclusive.

available knowledge and assumptions. CREAM accommodates alternative approaches under different assumptions about the reasoning graph, supporting modular designs aligned with available knowledge. Overall, it provides an interpretable, modular and adaptable framework balancing predictive accuracy with controlled reasoning.

## 2 RELATED WORK

**Connection to Neurosymbolic approaches.** CBMs are complementary to Neurosymbolic (NeSy) approaches (Badreddine et al., 2022; Bortolotti et al., 2024; Manhaeve et al., 2018; Marconato et al., 2023a); the former require supervision solely on the concepts while the latter require knowledge in the form of logical programs. In our case, we require knowledge about *(directed) statistical (in)dependencies* between interpretable variables, to constrain the relationships between them. In App. H we provide a logic-based viewpoint to CREAM.

**Concept and Task Relationships.** Standard CBMs assume that concepts are *independent* and that all of them directly contribute to the task, thus forming a bipartite graph ($G$). To address this, several works have incorporated concept interdependencies. Relational CBMs (Barbiero et al., 2024) use graph-structured data and message-passing algorithms to propagate relational dependencies, while Stochastic CBMs (SCBMs) (Vandenhirtz et al., 2024) model concepts using a learnable covariance matrix. Similarly, Autoregressive CBMs (ACBMs) (Havasi et al., 2022) introduce an autoregressive structure to learn sequential dependencies between concepts. These methods do not explicitly model expert-desired `C-C` and `C→Y` relationships. The closest to our work is Causal CGMs (Dominici et al., 2025), while concurrent work is $C^2$BMs (De Felice et al., 2025). The core differences lie in modeling and implementing $G$. The former learns relationships between endogenous variables and their copies, and embeds the concept representations in a higher-dimensional space. While the latter requires $G$ to be acyclic and assumes each relationship to be linear. Lastly, none of the prior works explicitly handle mutually exclusive concepts.

**Concept Incompleteness.** CBMs rely on predefined concept sets, causing lower accuracy when the concept set is incomplete (i.e., not a sufficient statistic for the target) (Mahinpei et al., 2021; Yeh et al., 2020; Zarlenga et al., 2022). For instance, in Fig. 1, a garment labeled as "Tops" may correspond to multiple classes, making exact classification impossible. Such cases arise when (i) concept annotation is costly, (ii) sparse explanations are preferred, or (iii) domain knowledge is limited. To address this issue, CBMs have been extended to incorporate side-channels. These hybrid CBMs have a lower upper bound of generalization error (Hayashi & Sawada, 2024) and capture unsupervised concepts (Sawada & Nakamura, 2022), residuals (Yuksekgonul et al., 2023; Zabounidis et al., 2023), or other auxiliary information (Dominici et al., 2025; Havasi et al., 2022). In contrast, we regularize the side-channel to prioritize concept importance.

**Concept Leakage.** Furthermore, CBMs suffer from concept leakage, a phenomenon tied to reasoning shortcuts (Bortolotti et al., 2024; 2025; Geirhos et al., 2020; Marconato et al., 2023b), where extra unintended information is encoded into concepts (Mahinpei et al., 2021; Makonnen et al., 2025; Marconato et al., 2023a; Margeloiu et al., 2021; Ragkousis & Parbhoo, 2024), leading to high accuracy even with irrelevant concepts and thus unreliable reasoning. Leakage has been suggested to be inherent in concept-based models that rely on concept embeddings (e.g., (De Felice et al., 2025; Dominici et al., 2025; Zarlenga et al., 2022)) challenging their interpretability (Parisini et al., 2025). Existing methods to mitigate leakage include using binary concept representations (Havasi et al.,

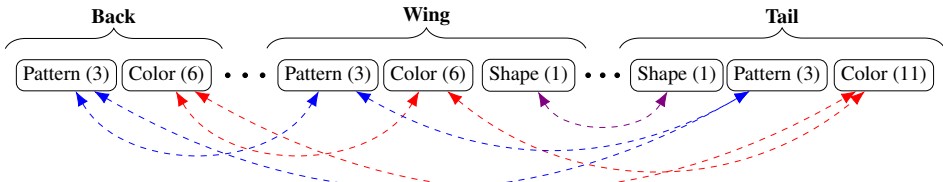

Figure 2: Partial illustration of CUB's reasoning graph. Concepts are represented as nodes, with numbers in parentheses indicating the cardinality for each concept. Nodes within the same group are mutually exclusive and disconnected, while edges are one-to-one between nodes from different groups. Bidirected edges indicate statistical dependencies between concepts.

2022; Lockhart et al., 2022; Sun et al., 2024; Vandenhirtz et al., 2024), training a CBM model in an independent manner (Margeloiu et al., 2021), using orthogonality losses (Sheth & Ebrahimi Kahou, 2023) or disentanglement techniques (Marconato et al., 2022; Sinha et al., 2024). Meanwhile, the reasoning structure of CREAM allows only for intended information flows, thus mitigating leakage by design, without needing hard concepts or introducing regularization.

# 3 CONCEPT REASONING MODEL

In this work, we propose *reasonable concept bottleneck models* that are guided, but not strictly limited, by the designer-picked or automatically discovered `C-C` and `C→Y` relationships. Unlike standard CBMs, which typically follow a bipartite concept-to-task architecture, we introduce CREAM as a framework that supports flexible, interpretable model reasoning while maintaining high performance.

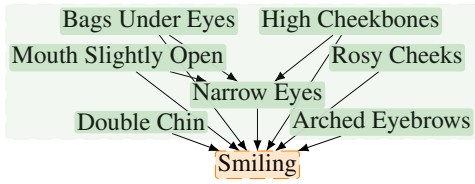

Figure 3: Reasoning graph for predicting "Smiling" in CelebA (Liu et al., 2015), showing the most correlated facial concepts ($C$) that *directly* influence the prediction ($Y$).

At the core of CREAM is the *model reasoning graph* $G = (V, E)$, which encodes the specified `C-C` and `C→Y` relationships. Formally, the node set is $V = C \cup Y$, and the edge set $E \subseteq V \times V$ captures plausible (un)directed relationships, representing information flow within the model.

To operationalize this structure, we partition $G$ into two subgraphs: the concept graph[1] $G_C \coloneqq G[C] \triangleq (C, E_{G_C})$ containing the `C-C` relationships, and the task graph $G_Y \coloneqq G[C_{direct} \cup Y] \triangleq (C_{direct} \cup Y, E_{G_Y})$ for `C→Y` reasoning. Here, $C_{direct} \coloneqq \{v \in C \mid \exists y \in Y : (v, y) \in E\}$ denotes the subset of concepts directly connected to $Y$.

## 3.1 CONCEPT-CONCEPT REASONING

Although CREAM is not restricted to categorical concepts, we assume them in this context and represent them as one-hot encoded vectors of length equal to number of categories. For example, in the concept graph $G_C$ of Fig. 2, the concept "Tail Color" has cardinality 11 and is thus represented as a vector of 11 mutually exclusive binary variables. Henceforth, we assume the concept set $C$ is in this binarized form, with $K \coloneqq |C|$ total concepts.

The concept graph's $G_C$ adjacency matrix, $A_C \in \{0, 1\}^{K \times K}$ is defined as follows:

$$A_C(i,j) = \begin{cases} 1 & \text{if } i = j \ \vee \ (c_i, c_j) \in E_{G_C}, \text{ for } c_i, c_j \in C; \\ 0 & \text{otherwise, i.e. undesired information flows} \end{cases} \tag{1}$$

**Types of `C-C` Relationships.** $A_C$ allows us to capture both *hierarchical relationships* and *correlations* between them, represented as asymmetric $(A_C(i,j) \neq A_C(j,i))$ and symmetric $(A_C(i,j) = A_C(j,i))$ entries respectively. For instance, in the graph of Fig. 3, all concept relationships are

---

[1]The graph can be disconnected, as seen in the seasonality concepts of cFMNIST, shown in Fig. 1.

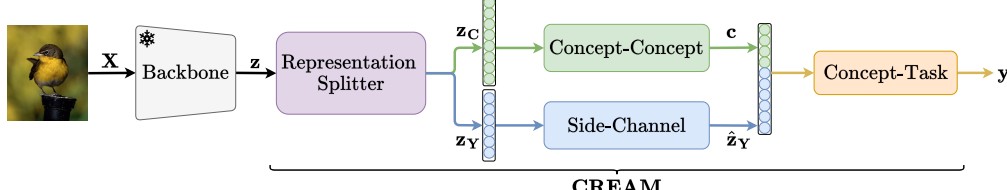

Figure 4: Sketch of CREAM's framework. The backbone's output is split into concept ($\mathbf{z}_C$) and side-channel ($\mathbf{z}_Y$) representations for concept and task prediction, respectively. The Concept-Concept block models relationships between concepts, while the Concept-Task block uses both $\hat{\mathbf{z}}_Y$ and the concepts to predict the task label based on embedded relationships.

hierarchical (e.g., "High Cheekbones" lead to "Narrow Eyes" but not vice versa). In contrast, Fig. 2, includes bidirected edges accounting for concepts that are correlated (such as wing colors). By combining the above, $A_C$ can also represent a Partially Directed Acyclic Graph (PDAG). An example of this is shown in App. E.5. Lastly, in Table 1 we show how the different concept-based models and the relationships they encode, can be implemented within CREAM's framework.

## 3.2 CONCEPT-TASK REASONING

Let $L := |Y|$ be the cardinality of the target variable $Y$. Following Section 3.1, in multiclass settings we assume the target variable has been binarized. We assume all C→Y edges are directed from concepts to task classes, reflecting experts reasoning "downwards" toward the targets.

Unlike prior CBMs, CREAM does not require all concepts to connect directly to the task. Indirect concepts, $C_{indirect} := C \setminus C_{direct}$ help predict other concepts within $G_C$, enhancing interpretability and enabling the intermediate steps of reasoning to be traced and verified. For instance, in Fig. 1, "Clothes" influences the final prediction indirectly through "Tops", which links to the target classes "T-Shirt", "Pullover", and "Shirt".

As before, we encode the C→Y relationships in $G_Y$ using a task adjacency matrix[2] $A_Y \in \{0,1\}^{K \times L}$

Table 1: Implementation of prior approaches' C−C relationships in CREAM. For CGM we leave it at $A_C$, as it best describes a PDAG.

| Model | Relationships | $A_C$ |
|---|---|---|
| CBM | Independent | $\mathbb{I}_K$ |
| ACBM | Autoregressive | $(\mathbb{1}_{i<j})_{i,j=1}^K$ |
| SCBM | Correlations | $A_C = A_C^T$ |
| C2BM | DAG | sparse $(\mathbb{1}_{i<j})_{i,j=1}^K$ |
| CGM | Causal graph | $A_C$ |

$$A_Y(i,j) = \begin{cases} 1 & \text{if } (c_i, y_j) \in E_{G_Y}, \text{ for } c_i \in C_{direct}, y_j \in Y; \\ 0 & \text{otherwise} \end{cases} \tag{2}$$

**Ease of Interventions.** Given a known $G$, users can identify errors and correct them via interventions easier. Since only $C_{direct}$ is used for predictions, the number of *effective interventions* is lowered from $K$ to $|C_{direct}|$. Also, humans can identify which concept predictions led to incorrect task predictions by tracing the edges ($G_Y$) and prioritize interventions on those specific concepts. Hence, $G_Y$ reduces human effort and complements existing concept selection criteria (Shin et al., 2023). Moreover, $G_C$ highlights relationships among concepts, such as mutually exclusive groups, facilitating grouped interventions rather than individual ones (Koh et al., 2020; Shin et al., 2023).

## 3.3 CONCEPT LEAKAGE

CBMs sometimes surpass the theoretical upper bound of concept-based task performance. This phenomenon, called *concept leakage*, is attributed to (i) concept representations inadvertently encoding extra information beyond their symbolic counterparts (Havasi et al., 2022; Mahinpei et al., 2021; Marconato et al., 2022; 2023a; Margeloiu et al., 2021; Sun et al., 2024; Parisini et al., 2025), and (ii) task predictions exploiting irrelevant concepts (Mahinpei et al., 2021; Sun et al., 2024). As a result, task predictions do not rely on learned concepts as intended, leading to erratic behavior under

---

[2]Although adjacency matrices are conventionally square, we refer to this $K \times L$ binary matrix as an adjacency matrix for notational consistency.

interventions and reduced interpretability. We define concept leakage as the surplus task accuracy of a model $f$ when using predicted concepts relative to the Bayes optimal predictor with true concepts: $\Lambda = \max(ACC_f - ACC_{Optimal}, 0)$ (Marconato et al., 2023a).[3] An exception occurs when models exploit side channels, since these can legitimately exceed the theoretical concept-based accuracy. Interventions on models with leakage may further reduce task accuracy, since human edits inject the exact intended information into the concepts (Margeloiu et al., 2021; Parisini et al., 2025). Our empirically validated hypothesis is that enforcing a structured reasoning process through $A_C$ and $A_Y$ constrains spurious or semantically invalid pathways, compelling the model to use intended relationships, thereby reducing leakage and improving interpretability.

## 4 DESIGNING CREAM

The CREAM framework, shown in Fig. 4, affords any CBM variant to be reformulated by embedding them with the following plug-and-play components: i) a *representation splitter* which decomposes the backbone feature representation ($\mathbf{z}$) into a concept representation ($\mathbf{z}_C$) and an optional *side-channel* representation ($\mathbf{z}_Y$); ii) a *concept-concept block* that enforces the C-C relationships via $A_C$; iii) a regularized side-channel; and iv) a *concept-task block* that encodes the C→Y reasoning via $A_Y$, and leverages the side-channel. We showcase their modularity and effects in Section 5.2 and App. E.2, E.4.

### 4.1 REPRESENTATION SPLITTER

CREAM builds atop a frozen pre-trained or fine-tuned backbone, which given an input image $X$ extracts a feature vector $\mathbf{z}$ serving as the initial information bottleneck. Then, a learnable *representation splitter* linearly partitions $\mathbf{z}$, into two disjoint latent representations:

1. **Concept exogenous variables** $\mathbf{z}_C \in \mathbb{R}^{d_C K}$ that serve as *input to the concept-concept block*, which enforces the C-C relationships. We assume a uniform latent capacity per concept for simplicity.

2. **Side-channel information** $\mathbf{z}_Y \in \mathbb{R}^{|\mathbf{z}| - d_C K}$ capturing information beyond the predefined concepts.

The dimensionalities of both $\mathbf{z}_C$ and $\mathbf{z}_Y$, are hyperparameters that influence the model's performance. To incorporate the reasoning structure encoded by $G$ and the functional (in)dependence constraints it implies, we draw inspiration from Structural Causal Models (SCMs) (Pearl, 2009). In an SCM, each endogenous variable $X_i$ (here, $X_i \in \mathbf{X} = C \cup Y$) is modelled as a function of its causal parents $\mathrm{pa}(X_i)$ (given by C-C and C→Y) and an exogenous noise variable $\mathbf{z}_i \in \mathbf{z}_C \cup \hat{\mathbf{z}}_Y$, i.e., $X_i = f_i(\mathrm{pa}(X_i), \mathbf{z}_i)$. Note that we do not aim to be causally consistent (i.e., we do not impose any causal assumptions, nor explicitly define the equations $f$), but use causality as a guiding analogy.

**Structured Neural Networks.** Structured Neural Networks (StrNNs) (Chen et al., 2024) enforce the functional (in)dependence constraints implied by $G$, meaning a variable $X_i$ must not be influenced by the exogenous noise of a $X_j$ that is not its parent ($\frac{\partial X_i}{\partial \mathbf{z}_j} = 0, \{\forall j \mid X_j \notin \mathrm{pa}(X_i)\}$). Given the number of hidden layers $d$, layer widths $(h_1, h_2, \ldots, h_d)$, and an adjacency matrix $\mathbf{A} \in \{0, 1\}^{p \times q}$ as hyperparameters, StrNN constructs a series of binary masks $M_1, \ldots, M_d$ which zero out non-permitted connections, ensuring the desired independencies are encoded while preserving maximal expressivity within those constraints. A detailed explanation of StrNNs can be found in App. A.

### 4.2 CONCEPT-CONCEPT BLOCK

This block enforces the C-C relationships encoded in $A_C$. To improve predictive performance, w.l.o.g., we assume each concept is associated with a $d_C$-dimensional exogenous embedding, yielding an input representation $\mathbf{z}_C \in \mathbb{R}^{d_C K}, d_C \in \mathbb{N}$. To enforce the concept graph $G_C$, we generate binary masks using StrNNs: $M_C := A_C^T \otimes \mathbb{1}_{1 \times d_C}$, where $\otimes$ denotes the Kronecker product. This ensures that each concept receives input only from the appropriate parents' exogenous vectors. The resulting concept-concept block $g : \mathbb{R}^{d_C K} \to \mathbb{R}^K$ receives as input $\mathbf{z}_C$ and relies on the concept mask $M_C$ to compute the *concept logits*, $\hat{l}_C \in \mathbb{R}^K$ as :

$$\hat{l}_{C_i} = g(\mathbf{z}_{C_i}, \mathrm{pa}(\mathbf{z}_C)), \quad \text{where } \mathrm{pa}(C_i) = \{v \in V | A[v, C_i] = 1\}. \quad (3)$$

---

[3]We adapt the definition in (Marconato et al., 2023a) from classification loss to accuracy.

This formulation follows a compacted SCM principle (Javaloy et al., 2024), whereby each concept depends only on its parents and its corresponding exogenous variables. Standard CBM lacks this structure and connects all $\mathbf{z}_C$ to every concept allowing for entangled and unwanted reasoning paths.

**(Mutex) Concept Representations.** The concept-concept block supports hard, soft, logit, and embedding representations. For mutually exclusive *(mutex)* concepts in $G_C$, we apply a softmax over the logits within each group. For non-mutex concepts, we apply the respective activations independently. In the main paper we focus on soft concepts, i.e., $\hat{C} := \sigma(\hat{l}_C)$, while an analysis of hard concepts in CREAM is provided in App. G.

## 4.3 SIDE-CHANNEL

The optional side-channel information $\mathbf{z}_Y$ is projected by an MLP to $\hat{\mathbf{z}}_Y \in \mathbb{R}^L$ and *serves as the exogenous input to the tasks*, assigning each class its own exogenous variable. W.l.o.g., we assume each class needs exactly one exogenous variable for prediction. Increasing the dimensionality of $\hat{\mathbf{z}}_Y$ may improve performance but reduces concept importance. We empirically show in App. E.4.1 that the side-channel in CREAM primarily supports classes that cannot be predicted from concepts alone.

**Regularization of side-channel.** Adding a black-box side-channel to CBMs boosts task performance, but may reduce interpretability, since task predictions can use non-interpretable predictors. To control this, we apply a dropout-based regularization (Huang et al., 2016), dropping the entire side-channel with probability $p$. This encourages the model to favor concepts, using the side-channel only when needed. At inference, the side-channel can be dropped for purely concept-based predictions.

## 4.4 CONCEPT-TASK CLASSIFIER

The final stage of CREAM maps concept predictions to task logits while incorporating side-channel in a controlled manner. Similar to the concept-concept block, we use StrNN to enforce C→Y relationships expressed by $A_Y$. To incorporate the side-channel representation $\mathbf{z}_Y$, we parameterize the concept-task StrNN using the binary mask $M_Y := [A_Y^T; I_L]$, where $I_L$ denotes the identity matrix of size $L$ that connects each element in the side-channel representation $\mathbf{z}_Y$ with only one of the tasks classes. This ensures each class is dependent only on its parent concepts (Eq. 3) and, the optional class-specific latent features from the side-channel, leading to *sparser explanations*. Formally, the task prediction for class $j$, using a classifier $f$, usually a single layer MLP for interpretability, is computed as:

$$\hat{y}_j = f([\hat{c}_{Pa_j}, \mathbf{z}_{Y_j}]) \quad \text{where } f : \mathbb{R}^{K+L} \to \mathbb{R}^L, \hat{c}_{Pa_j} \subseteq C_{direct}. \tag{4}$$

## 4.5 TRAINING

To train CREAM, we adopt the joint bottleneck training scheme (Koh et al., 2020), which optimizes both the task loss ($\mathcal{L}_Y$) and the concept loss ($\mathcal{L}_C$) simultaneously, through linear scalarization. The optimization objective is to minimize the weighted sum of these losses ($\mathcal{L}$), for the observed training samples $\{(x^{(n)}, y^{(n)}, c^{(n)})\}_{n=1}^N$, where $x \in \mathbb{R}^{|\mathbf{z}|}$ is image embedding, $c \in \mathbb{R}^K$ are concepts, $y \in \{0, 1\}^L$ is the target class, and $\lambda > 0$ is the weight of concept compared to task performance:

$$\mathcal{L} = \sum_n \mathcal{L}_Y(\hat{y}^{(n)}; y^{(n)}) + \lambda \sum_n \sum_k \mathcal{L}_{C_k}(\hat{c}^{(n)}; c^{(n)}). \tag{5}$$

## 5 EXPERIMENTS

We evaluate our framework across standard datasets, demonstrating its ability to achieve the following desiderata. Our experiments assess: (i) concept and task accuracies, (ii) computational efficiency, (iii) intervenability, (iv) mitigation of concept leakage, and (v) the effect of the dropout regularization.

## 5.1 SETUP

**Datasets.** We evaluate CREAM on three image datasets selected for their distinct relational structures. **FashionMNIST** (Xiao et al., 2017) exhibits hierarchical and mutex relations with concept

Table 2: Task and concept accuracy(%). Reported values are mean and standard deviation. CREAM achieves a better balance between performance and interpretability. The best method is **bold** and the second-best is underlined. Relative training speed and peak memory are averaged across all datasets.

| Model | iFMNIST | | cFMNIST | | CUB | | CelebA | | All Datasets | |
|---|---|---|---|---|---|---|---|---|---|---|
| | $ACC_Y$ | $ACC_C$ | $ACC_Y$ | $ACC_C$ | $ACC_Y$ | $ACC_C$ | $ACC_Y$ | $ACC_C$ | Time | Peak Memory |
| **Black-box** | $92.70_{0.07}$ | - | $92.70_{0.07}$ | - | $74.85_{0.12}$ | - | $78.47_{0.30}$ | - | - | - |
| $C_{true} \to Y$ | $60.00$ | - | $100.00$ | - | $100.00$ | - | $84.51$ | - | - | - |
| **CBM** | $91.19_{0.40}$ | $96.74_{0.36}$ | $92.00_{0.17}$ | $97.33_{0.06}$ | $\mathbf{73.76_{0.32}}$ | $80.90_{0.08}$ | $79.28_{0.39}$ | $79.77_{0.17}$ | $\text{x}1.00_{0.00}$ | $\text{x}1.00_{0.00}$ |
| **ACBM** | $52.25_{5.18}$ | $98.94_{0.01}$ | $90.68_{0.09}$ | $98.03_{0.02}$ | $66.98_{0.43}$ | $\mathbf{94.15_{0.01}}$ | $81.40_{0.46}$ | $80.57_{0.53}$ | $\text{x}7.65_{4.08}$ | $\underline{\text{x}1.10_{0.09}}$ |
| **SCBM** | $57.68_{0.63}$ | $98.86_{0.02}$ | $90.80_{0.17}$ | $97.54_{0.06}$ | $70.55_{0.19}$ | $\underline{90.28_{0.04}}$ | $76.63_{0.47}$ | $80.54_{0.11}$ | $\text{x}17.00_{10.96}$ | $\text{x}1.12_{0.11}$ |
| **CGM$_{CD}$** | $68.81_{14.65}$ | $90.05_{5.28}$ | $67.92_{7.37}$ | $90.48_{1.20}$ | - | - | $\mathbf{81.70_{0.83}}$ | $\mathbf{82.17_{0.42}}$ | $\text{x}10.53_{7.42}$ | $\text{x}527.89_{447.06}$ |
| **CGM$_{prior}$** | $90.67_{0.21}$ | $98.77_{0.04}$ | $90.33_{0.19}$ | $97.72_{0.08}$ | - | - | $\underline{81.51_{1.36}}$ | $\underline{81.99_{0.13}}$ | $\text{x}6.39_{2.94}$ | $\text{x}31.80_{9.49}$ |
| **C$^2$BM** | $\underline{91.96_{0.18}}$ | $\underline{98.96_{0.02}}$ | $\underline{92.04_{0.23}}$ | $\underline{98.07_{0.02}}$ | - | - | $76.19_{1.17}$ | $78.93_{0.58}$ | $\text{x}10.06_{5.24}$ | $\text{x}1.32_{0.44}$ |
| **CREAM** | $\mathbf{92.43_{0.23}}$ | $\mathbf{99.07_{0.03}}$ | $\mathbf{92.38_{0.16}}$ | $\mathbf{98.08_{0.06}}$ | $\underline{72.90_{0.28}}$ | $86.83_{0.04}$ | $80.92_{0.55}$ | $79.91_{0.22}$ | $\mathbf{\text{x}1.81_{0.52}}$ | $\text{x}1.00_{0.00}$ |

incompleteness. We use two variants: *iFMNIST* with $K = 8$ hierarchical categories (Seo & Shin, 2019), and *cFMNIST* with $K = 11$ by adding seasonal attributes (Fig. 1). **CUB** (Wah et al., 2011) provides 112 correlated and mutex concepts describing fine-grained attributes such as tail color and wing pattern (Koh et al., 2020). **CelebA** (Liu et al., 2015) involves a DAG structure over seven facial attributes used to predict smiling (Fig. 3). Full dataset details are described in App. C.

**Model Baselines.** To establish references for task performance, we train a black-box model and models trained from the ground-truth concepts ($C_{true} \to Y$). For hard concept representations, we include ACBM,[4] and SCBM, using only the Amortized SCBM since it consistently outperforms the Global SCBM. We also evaluate embedding-based models: namely C$^2$BM, CGM$_{CD}$ that discovers a graph and its version that embeds a given graph (CGM$_{prior}$). Finally, we include a standard CBM, and tried including ECBM butwere unable to reproduce satisfactory results with our backbone.

**Implementation Details.** All models are initialized from a shared backbone network, which is fine-tuned for a few epochs on the respective datasets and then frozen to ensure consistent feature extraction. For each dataset, we perform hyperparameter tuning; detailed hyperparameter configurations are provided in App. D.[5] The construction of masking pathways in StrNNs follows the algorithm found in Zuko (Rozet et al., 2022). For all baseline models we adopt their proposed experimental settings.

## 5.2 KEY FINDINGS

**Bridging the gap between interpretability and performance.** Our results in Table 2 show that CREAM achieves competitive task and concept performance, even in incomplete settings, outperforming concept-based baselines and black-box models. Although CREAM's concept accuracy drops in CUB, its task performance remains competitive. Notably, CGM models are too slow for large graphs like CUB, due to their graph operations, and C$^2$BM is incompatible with CUB's reasoning. The two main components of CREAM offer orthogonal benefits: structured reasoning promotes interpretability, while the side-channel addresses limitations of incomplete or noisy concepts. We also report the training computational efficiency of all models, including their relative slowdown and memory requirements compared to the standard CBM model, across all datasets and runs. Note that both CGMs can only run on CPU; thus, we compare them to CBM on CPU. Our results show that, among all models, CREAM achieves the best computational efficiency, being both the fastest and requiring the least memory. Detailed per-dataset results can be found in App. E.1. In App. F, we study the effect of different hyperparameters on CREAM's performance.

**Intervenability.** We also assess task accuracy after intervening. For models with hard concepts, interventions are straightforward: the true concept values are directly inserted. However, for soft concept models, we set the concept activations to the 5th and 95th percentiles proposed in (Koh et al., 2020). For all models, we follow a random concept selection policy (Shin et al., 2023). However, causal models differ in that they avoid intervening on both parent and child concepts in the graph. For CREAM we randomly select from $C_{direct}$, the only concepts used for prediction. This sets an upper bound on the number of interventions (6 for iFMNIST, 9 for cFMNIST), reaching peak accuracy faster. For comparison, we also show the interventions on $C_{indirect}$ for CREAM. In CUB and CelebA, all concepts are direct, thus the number of effective interventions is 112 and 7, respectively.

---

[4]We use the implementation available in SCBM's repository.

[5]We will release our code publicly upon acceptance to ensure full reproducibility.

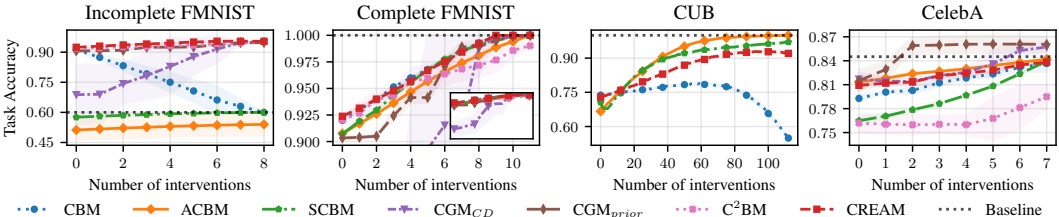

Figure 5: Impact of individual interventions on task accuracy. The baseline model is $C_{true} \to Y$. CREAM's accuracy improves with increasing number of interventions up to the number of $C_{direct}$. For cFMNIST, the inset axes show a zoomed-out view, to account for $CGM_{CD}$.

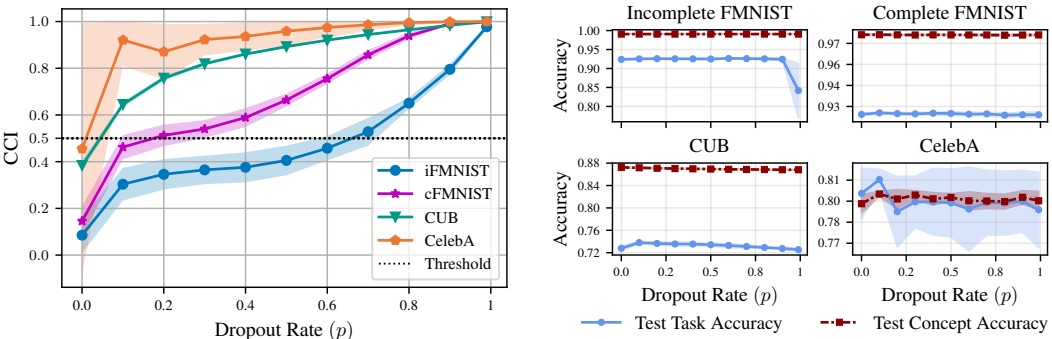

Figure 6: **Left:** Impact of dropout rates on concept channel importance. Models with complete concept sets need less dropout. **Right:** Concept and task vs dropout rate ($p$). Values are presented as mean $\pm$ standard deviation across 5 seeds. Increased $p$ leads to marginal drops in task accuracy.

Fig. 5 shows task accuracy across models after interventions. In iFMNIST, all models but CBM improve with more interventions, while CBM drops to the $C_{true} \to Y$ model's performance, indicating a leaky model (Margeloiu et al., 2021). Even after full interventions, no model reaches 100% In cFMNIST, accuracy generally rises with interventions, except for $C^2$BM, which declines after 5 interventions. Similarly, in CUB, most models improve, but soft CBM and SCBM drop under certain interventions. Also, almost no model reaches $C_{true} \to Y$, likely due to noisy concepts. As noted, $C^2$BM and CGM do not work in CUB. Lastly, in CelebA, CREAM matches the baseline after full interventions. CGM again surpasses the $C_{true} \to Y$ performance, at the cost of interpretability, while $C^2$BM initially maintains the same accuracy, but improves after multiple interventions. App. E.3, studies group intervention efficiency on mutex concepts for CREAM.

**Interpretability in the presence of a side-channel.** Since CREAM incorporates a black-box side-channel, verifying that predictions rely mainly on concepts is crucial. We introduce Concept Channel Importance (CCI), adapted from SAGE values (Covert et al., 2020) to quantify this. They measure *global feature importance* (Molnar, 2025) and conditional mutual information when used with an optimal model. CCI is defined as the normalized importance of the concept channel relative to total predictive capacity: $CCI = \frac{\phi_c}{\phi_c + \phi_y}$, where $\phi_c$ and $\phi_y$ denote the SAGE values of the whole concept and side channels, respectively. Values near 1 indicate more substantial reliance on concepts, and thus higher interpretability. An analysis of CCI and permutation feature importance (Breiman, 2001; Fisher et al., 2019) can be found in App. B. We show that $CCI > 0.5$ is enough for our desiderata to hold.

**Effect of Dropout** Fig. 6 shows that increasing the dropout rate $p$ raises CCI, promoting concept-based reasoning. Also, the need for side-channel regularization decreases when using complete concept sets. Importantly, models that use almost zero regularization fall below the CCI threshold, highlighting the importance of side-channel regularization that prior works ignored. These findings lead to a key conclusion: *dropout rate controls interpretability*. We also observe that increasing $p$ slightly reduces task performance in complete cases. Surprisingly, even with an extreme dropout rate of $p \approx 1$, CREAM maintains black-box level accuracy in incomplete datasets.

Table 3: CREAM's performance on iFMNIST without the side-channel. Leakage is avoided when using the `C→Y` relationships, while `C−C` relationships help mitigate it.

Figure 7: Correlation matrix of the exogenous variables **z**. The enforced reasoning is mirrored in CREAM.

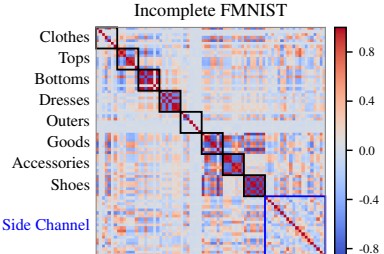

| Mutex | Reasoning | $ACC_Y$ | $ACC_C$ | $\Lambda$ |
|-------|-----------|---------|---------|-----------|
| No | `C−C` | $90.28_{4.2}$ | $96.38_{0.8}$ | $30.28_{4.2}$ |
| | `C→Y` | $57.31_{0.3}$ | $99.06_{0.0}$ | $0$ |
| | `C−C, C→Y` | $57.41_{0.6}$ | $99.04_{0.0}$ | $0$ |
| Yes | `C−C` | $67.60_{7.8}$ | $98.53_{1.1}$ | $7.60_{7.8}$ |
| | `C→Y` | $57.32_{0.2}$ | $99.05_{0.0}$ | $0$ |
| | `C−C, C→Y` | $57.10_{0.1}$ | $99.07_{0.0}$ | $0$ |

**Concept Leakage.** We evaluate whether CREAM suffers from concept leakage. In Section 3.3 we defined concept leakage as $\Lambda = \max(ACC_f - ACC_{Optimal}, 0)$, i.e., the model should not outperform the $C_{true} \to Y$ baseline *when using only concepts*. We focus on iFMNIST because it is the only dataset where the $C_{true} \to Y$ baseline is outperformed; and soft-concept models are prone to leakage in incomplete concept settings (Mahinpei et al., 2021; Parisini et al., 2025). For instance, CBM is leaking in iFMNIST: as shown in Table 2, since it exceeds the $C_{true} \to Y$ model.

By stripping away CREAM's plug-and-play components, we empirically show that its structured reasoning helps mitigate leakage. To isolate the mechanisms responsible, we *remove the side-channel* and then: (i) remove `C−C` reasoning (ii) remove `C→Y` reasoning, and (iii) replace softmax with sigmoid to treat concepts as independent. Table 3 shows that CREAM avoids leakage despite being a soft model. Specifically, `C→Y` reasoning entirely prevents it, while `C−C` relationships help mitigate it. Notably, softmax enforces mutual exclusivity, reducing leakage, whereas sigmoid allows for more. Fig. 7 confirms that correlations among exogenous variables reflect the imposed relationships, e.g., {"Goods","Accessories","Shoes"} belong to the same sub-tree, are highly correlated. Also, the $\mathbf{z}_C$ variables of a concept are highly correlated with each other. A detailed analysis can be found in App. E.4.

# 6 CONCLUSION AND FUTURE WORK

In this work, we introduced CREAM, a computationally efficient and flexible CBM framework that enables experts to encode prior knowledge about `C−C` and `C→Y` relationships into model reasoning. Its modular design supports diverse `C−C` relationships and concept representations. It narrows the interpretability-performance gap, especially in concept-incomplete settings, through a regularized side-channel, and facilitates interventions and interpretability via sparser `C→Y` reasoning. Importantly, we showed that proper regularization of the side-channel (e.g., via dropout) is crucial to maintaining interpretability. To further evaluate models in this setting, we introduced $CCI$, a new metric for quantifying interpretability when predictions rely on auxiliary channels beyond the concept channel. Empirically, we demonstrated that CREAM's structured reasoning effectively avoids concept leakage, making it, to the best of our knowledge, the first CBM framework that is leakage-free while operating with soft concepts.

**Future Work and Limitations.** CREAM requires prior domain knowledge to encode concept-concept and concept-task relationships. Future work could explore automated structure learning (Zanga et al., 2022) to infer these dependencies, as shown in App. E.5. Additionally, implementing adaptive dropout strategies, where the side-channel is dynamically leveraged based on concept prediction uncertainty, could improve robustness and interpretability while reducing the effort required for hyperparameter tuning. The side-channel can also be used to discover new concepts (Sawada & Nakamura, 2022) that are not present in the concept bottleneck.

# 7 REPRODUCIBILITY STATEMENT

We ensure reproducibility of our work by properly crediting and respecting the licenses of all datasets and assets used, with full citations and links provided in Section 5.1 and Appendix D. The experimental setup, including training and testing details such as data splits, hyperparameters,

and optimizers, is explicitly described in Section 5.1 and Appendices C and D. We provide open access to our code in the supplementary materials, along with instructions to reproduce the main results, and will make both code and data publicly available upon acceptance. To facilitate faithful replication, we disclose the hyperparameters of all methods and detail the computational resources used in Appendix D. In addition, we report the type of compute workers, memory, and execution time required for our experiments in Appendix D and E.1. Together, these measures ensure that all main experimental results and conclusions of the paper can be independently verified.

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

# A  STRUCTURED NEURAL NETWORKS FOR CONCEPT-CONCEPT AND CONCEPT-TASK RELATIONSHIPS

Structured Neural Networks (StrNN) (Chen et al., 2024) enforce functional independence between inputs and outputs using masking pathways that preserve the structural constraints dictated by an adjacency matrix. Given a function $f : \mathbb{R}^m \rightarrow \mathbb{R}^n$, where $z \in \mathbb{R}^m$ is the input and $\hat{z} \in \mathbb{R}^n$ is the predicted output, StrNN ensures that dependencies between inputs and outputs adhere to a given adjacency matrix $A \in \{0,1\}^{m \times n}$. This is enforced through the condition:

$$A_{ij} = 0 \implies \frac{\partial \hat{z}_j}{\partial z_i} = 0. \tag{6}$$

This means that if $A_{ij} = 0$, the output $\hat{z}_j$ remains independent of input $z_i$, maintaining the prescribed reasoning structure of the $G$.

**Masks in Structured Neural Networks**  For both concept and task prediction networks, layer-wise masking pathways are applied to enforce structured reasoning $A^{m \times n}$. Given a network with $d$ hidden layers, each with widths $h_1, h_2, \ldots, h_d$, we define binary masks:

$$M_1 \in \{0,1\}^{h_1 \times m}, \quad M_2 \in \{0,1\}^{h_2 \times h_1}, \quad \ldots, \quad M_d \in \{0,1\}^{n \times h_d}, \tag{7}$$

such that:

$$M' = M_d \cdot \ldots \cdot M_2 \cdot M_1 \approx M, \tag{8}$$

where $M' \in \{0,1\}^{n \times m}$ maintains the same sparsity pattern as $A^T$, ensuring structured dependencies are preserved. The structured neural network function $\text{StrNN}_M$ is defined as:

$$\hat{z} = \text{StrNN}_M(z) = f_{d+1}\left(\ldots f_1\left(\left(W_C \odot M_1\right) z + b_C\right)\right), \tag{9}$$

where each layer transformation follows:

$$f_i(z) = a\left(\left(W_{C(i)} \odot M_i\right) z + b_i\right), \quad \forall i \in \{1, \ldots, d\}, \tag{10}$$

where: $W_{C(i)} \in \mathbb{R}^{h_i \times h_{i-1}}$ is the learnable weight matrix at layer $i$, $M_i \in \{0,1\}^{h_i \times h_{i-1}}$ is the binary mask ensuring structured dependencies, $b_i \in \mathbb{R}^{h_i}$ is the bias term and $a(\cdot)$ is the activation function.

For the concept-concept learning block $\text{StrNN}_{M_C}$:

$$m = d_C K, \quad n = K, \quad h_{i \geq 1} = d_C K. \tag{11}$$

For the classifier $\text{StrNN}_{M_Y}$:

$$m = K + L, \quad n = L, \quad h_{i \geq 1} = K + L. \tag{12}$$

The depth $d$ is treated as a hyperparameter. By enforcing structured dependencies, StrNN ensures that both concept and task predictions follow expert-defined reasoning pathways, enhancing interpretability without sacrificing predictive performance.

## A.1  CONCEPT-CONCEPT MASKING

The adjacency matrix for concept relationships is given by $A_C \in \{0,1\}^{K \times K}$, as defined in Section 3.1. The input to the Concept-Concept block is obtained from the representation splitter and is denoted as:

$$\mathbf{z}_C = (z_1, z_2, \ldots, z_{d_C K}) \in \mathbb{R}^{d_C K}. \tag{13}$$

Note that each dimension in $\mathbf{z}_C$ is a result of an expansion; each exogenous variable was duplicated from dimensionality of 1 to dimensionality of $d_C$. This operation is represented by the Kronecker product ($\otimes$). These extra dimensions must still follow the independencies of the original variable. Thus, we construct the concept mask $M_C \in \{0,1\}^{K \times d_C K}$ by:

$$M_C = A_C^T \otimes \mathbb{1}_{1 \times d_C}, \tag{14}$$

where $\mathbb{1}_{1 \times d_C}$ is a row vector of ones that replicates $A_C^T$ column-wise, ensuring structured reasoning of concept-concept relationships across all feature dimensions.

## A.2 CONCEPT-TASK MASKING

For the concept-task classifier (Section 4.4), the input combines both concept predictions and the side-channel information. Thus, its input is the concatenated $\hat{C}$ and $\hat{\mathbf{z}}_Y$, and thus it is of size $\mathbb{R}^{K+L}$. The C→Y relationships are described in the adjacency matrix $A_Y \in \{0,1\}^{K \times L}$. Meanwhile, since we assign one dimension of side-channel variables ($\hat{\mathbf{z}}_Y$) to each task, we will need to expand the adjacency matrix used in StrNN, using an identity matrix $I_L$. From this, we can define the concept-task mask as:

$$M_Y = \left[ A_Y^T ; I_L \right]. \tag{15}$$

This ensures that each class prediction depends only on the relevant parent concepts and its assigned side-channel node.

## B FEATURE IMPORTANCE METRICS

Given that CREAM integrates a side-channel that can contribute to task predictions, it is crucial to assess the relative importance of the concept set $C$ compared to it. To quantify this, we employ two model-agnostic metrics: Concept Channel Importance ($CCI$) and Permutation Feature Importance ($PFI$). These two metrics collectively provide an assessment of whether CREAM effectively balances interpretability and performance by ensuring that predictions remain grounded in human-understandable concepts rather than being dominated by the side-channel. In Table 4, we report the values of these importance metrics, for the models used in Section 5.

### B.1 CONCEPT CHANNEL IMPORTANCE

Concept Channel Importance ($CCI$) is based on Shapley Additive Global Explanations (SAGE) (Covert et al., 2020), which provide model-agnostic feature importance scores. Specifically, the SAGE value $\phi_i(v_f)$ of feature $i$, represents the Shapley values (Shapley et al., 1953) for the cooperative game $v_f(S)$. The cooperative game $v_f$ represents the expectation of the per-instance reduction in risk when using a subset of features $S \subseteq D$:

$$v_f(S) = \mathbb{E}[\mathcal{L}(f_\varnothing(X_\varnothing), Y)] - \mathbb{E}[\mathcal{L}(f_S(X_S), Y)],$$

where $f_\varnothing(x_\varnothing)$ is the model prediction without using any features, (i.e., the mean model prediction $\mathbb{E}[f(X)]$), and $f_S(X_S)$ the prediction using only the subset of features $S$ of all features $D$ ($S \subseteq D$). In our case, $\mathcal{L}$ is given by Equation 5, $f$ is the concept-task classifier and $D = [\hat{C}, \mathbf{z}_Y]$. Given $v_f(S)$, SAGE is then calculated by:

$$\phi_i(v_f) = \frac{1}{|D|} \sum_{S \subseteq D \setminus \{i\}} \binom{|D|-1}{|S|}^{-1} \Big( v_f(S \cup \{i\}) - v_f(S) \Big).$$

SAGE values provide global interpretability (Molnar, 2025) instead of explanations for individual predictions. The features that the model deems most useful will have positive SAGE values, non informative features have values close to zero, and harmful for the prediction features have negative values. Lastly, SAGE values can also measure a weighted average of conditional mutual information when they are used with an optimal model trained with the cross entropy or MSE loss. Specifically, the SAGE value of a feature $i$ used in optimal model $f^*$, is equal to:

$$\phi_i(v_{f^*}) = \frac{1}{|D|} \sum_{S \subseteq D \setminus \{i\}} \binom{|D|-1}{|S|}^{-1} I(Y; X_i | X_S).$$

We also briefly mention some of the properties that SAGE satisfies:

- **Efficiency:** SAGE values sum up to the total predictive power of all of the features (SAGE value using all the features $D$):. $\sum_{i=1}^{K} \phi_i(v_f) = v_f(D)$.
- **Dummy:** If a feature makes zero contribution, i.e., if it is an uninformative feature, then $\phi_i(v_f) = 0$.

Table 4: Interpretability metrics: CREAM variants (S-CREAM for soft concepts, H-CREAM for hard concepts) show higher concept-channel importance relative to the side-channel's across all datasets. Also, S-CREAM exhibits larger CCI values compared to H-CREAM in FMNIST.

| Dataset | Model | $CCI \uparrow$ | $PCI \uparrow$ | $PSI \downarrow$ |
|---|---|---|---|---|
| iFMNIST | H-CREAM | $0.80_{0.01}$ | $0.66_{0.06}$ | $0.35_{0.00}$ |
| | S-CREAM | $0.80_{0.02}$ | $0.59_{0.06}$ | $0.35_{0.00}$ |
| cFMNIST | H-CREAM | $0.88_{0.02}$ | $0.66_{0.03}$ | $0.07_{0.04}$ |
| | S-CREAM | $0.94_{0.02}$ | $0.72_{0.03}$ | $0.03_{0.01}$ |
| CUB | S-CREAM | $0.96_{0.00}$ | $0.72_{0.00}$ | $0.01_{0.00}$ |
| CelebA | S-CREAM | $0.92_{0.11}$ | $0.30_{0.02}$ | $0.01_{0.01}$ |

Interestingly, SAGE values can be generalized to group of features. Since our objective is to measure the overall contribution of the concept channel, we treat all of the concepts as one coalition. The SAGE value of a coalition represents how much this group of features improves the model's predictive ability.

Concept channel importance ($CCI$) can also be expressed is terms of total predictive power $v_f(D)$:

$$CCI = \frac{\phi_c(v_f)}{\phi_c(v_f) + \phi_y(v_f)} \overset{\text{Efficiency}}{=} \frac{\phi_c(v_f)}{v_f(D)}$$

where $\phi_c(v_f), \phi_y(v_f)$ denote the SAGE value of the whole concept-channel and side-channel respectively.

### B.1.1 DERIVING THE DESIRED IMPORTANCE THRESHOLD

We desire the importance of the concept channel to be greater or equal to the importance of the side channel, i.e., $\phi_c(v_f) > \phi_y(v_f)$. Assuming that *both channels are informative* (i.e., $\phi_c(v_f), \phi_y(v_f) > 0$), we derive a desired lower threshold for the concept channel importance:

$$\phi_c(v_f) \geq \phi_y(v_f) \qquad \text{add } \phi_c(v_f) \text{ to both sides}$$
$$\iff 2\phi_c(v_f) \geq \phi_y(v_f) + \phi_c(v_f) \qquad \text{assuming } \phi_c(v_f), \phi_y(v_f) > 0$$
$$\iff \frac{1}{2\phi_c(v_f)} \leq \frac{1}{\phi_y(v_f) + \phi_c(v_f)} \qquad \text{multiply both sides with } \phi_c(v_f)$$
$$\iff \frac{\cancel{\phi_c(v_f)}}{2\cancel{\phi_c(v_f)}} \leq \frac{\phi_c(v_f)}{\phi_y(v_f) + \phi_c(v_f)}$$
$$\iff \frac{1}{2} \leq CCI$$

Meanwhile, if the side channel is uninformative, then due to the dummy property $\phi_y(v_f) = 0$, then $CCI = 1$. For the sake of completion, in the edge case where the side-channel makes the prediction less accurate, i.e., $\phi_y(v_f) < 0$, then $CCI \in (-\infty, 0) \cup (1, +\infty)$. Note that we do not notice such cases in our experiments. Lastly, $CCI = 0$, if the concepts are not used by the Concept-Task Block.

In conclusion, assuming that both SAGE values are positive, $CCI$ is bounded between $CCI \in [0, 1]$. A $CCI$ value close to 1 indicates that the model relies primarily on the concept channel, reinforcing interpretability. When $CCI \approx 0.5$, it suggests that both the concept and side-channels contribute equally to the predictions. To compute CCI, we evaluate the trained model on the entire test set, storing the classifier's inputs to estimate feature attributions.

## B.2 PERMUTATION FEATURE IMPORTANCE

Permutation Feature Importance ($PFI$) (Breiman, 2001; Fisher et al., 2019) measures the significance of a feature by evaluating how much the model's accuracy deteriorates when its values are randomly shuffled. A feature is considered important if permuting its values leads to a significant drop in accuracy, whereas an unimportant feature results in little to no change. The PFI score for a feature $j$ is computed as:

$$PFI_j = ACC_Y - \frac{1}{K} \sum_{k=1}^{K} ACC_{Y_{j,k}}$$ 

(16)

where $ACC_Y$ is the test accuracy of the model, and $ACC_{Y_{j,k}}$ is the accuracy after randomly permuting the values of feature $j$ for the $k$-th iteration. In our case, we focus on evaluating the importance of entire channels (i.e., groups of features). Thus we permute all values within each channel (concept channel and side-channel) simultaneously. We denote concepts' feature importance as **P**ermutation **C**oncept **I**mportance (PCI) and the side-channel's as **P**ermutation **S**ide-Channel **I**mportance (PSI). We measure PFI on the test dataset (Molnar, 2025), and we permute for 100 iterations. Lastly, as mentioned, we desire the concept-channel to be more important than the side-channel, i.e., $PCI > PSI$. This inequality plays the same role as the importance threshold $CCI > 0.5$ derived in B.1.1. As seen in Table 4, the PFI-based metrics also indicate that CREAM prioritizes the concept-channel instead of the side-channel.

## B.3 DROPOUT RATE AND PFI

In Fig. 8, we present the Permutation Feature Importance curves for all datasets and models , when trained with different dropout rates ($p$). For each $p$, we train a model with it, and then we plot its $PCI$ and $PFI$, showing how much does the test accuracy drop if we randomly permute the concept-concept and side-channel, respectively. As expected, the findings are consistent with the observed trends in Section 5.2. Increasing the dropout rate $p$ leads to increased PCI and decreased PSI. Also, the need for side-channel regularization decreases when using complete concept sets. The point where $PCI > PSI$, in iFMNIST, is around $p = 0.75$, but when moving to the complete FMNIST case it drops to around $p = 0.3$. One difference between the PFI metrics and CCI, is the lowest required dropout rate $p$ such that the side-channel stops dominating the concept-concept block (i.e., $CCI > 0.5$ or $PCI > PSI$). Across all datasets, the PFI-based approach suggests that we should regularize the side-channel more than CCI suggests.

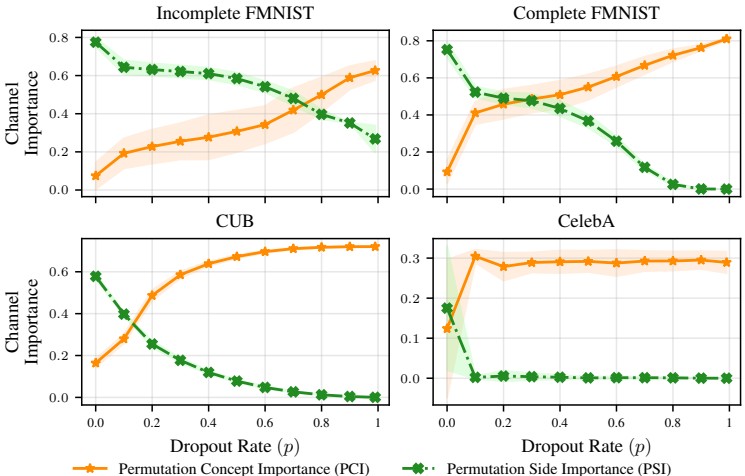

Figure 8: Mean values $\pm$ standard deviation of the permutation feature importance of both channels, in all datasets for CREAM, averaged over 5 seeds. Increasing $p$ leads to an increase in PCI and decreases PSI.

Table 5: Summary of each dataset. A concept group consists of a mutually exclusive group of concepts. $C_{direct}$ refers to the number of concepts directly connected to any task, which coincides with the upper bound of interventions. The reported number of effective group interventions refers to the number of directly connected groups of concepts instead.

| Dataset | Number of Classes ($L$) | Number of Concepts ($K$) | $C_{direct}$ | Mutex Groups | Effective Group Interventions |
|---|---|---|---|---|---|
| iFMNIST | 10 | 8 | 6 | 2 | 1 |
| cFMNIST | 10 | 11 | 9 | 3 | 2 |
| CUB | 200 | 112 | 112 | 27 | 27 |
| CelebA | 2 | 7 | 7 | 7 | 7 |

## C  DATASET DETAILS

A summary of the datasets used in our experiments is provided in Table 5.

### C.1  DATASET DESCRIPTION

**Apparel Classification (FashionMNIST)**  The FashionMNIST (FMNIST) [6] dataset (Xiao et al., 2017) consists of 70,000 grayscale images of clothing items across 10 classes. However, FMNIST does not provide predefined concept annotations or a dependency graph. To define concepts, we use high-level apparel categories derived from the hierarchical structure in (Seo & Shin, 2019). The resulting dependency graph $G$ forms a hierarchical tree, as shown in Figure 1.

We call this dataset Incomplete FMNIST. Since in hierarchical classification the classes at each level are mutually exclusive, the concepts of the same depth of the tree are also mutually exclusive. This means that the mutex concepts are grouped as such: {Clothes, Goods} and {Tops, Bottoms, Dresses, Outers, Accessories, Shoes}, leading to two groups of concepts. Note that, in the hierarchical classification setting the second group would be split into: {Tops, Bottoms, Dresses, Outers} and {Accessories, Shoes}. However, if we applied that logic to our case, both concept groups can be active at any time, thus being one mutex group. Lastly, these concepts cannot fully predict the classes. For instance, the same "active" concept vector $c = $ {Clothes, Tops} is used to predict these three classes $y =$ {T-shirt, Pullover, Shirt} with no way of distinguishing between them. The same problem applies for $c =$ {Goods, Shoes} and the classes $y = $ {Sandal, Sneaker, Ankle Boot}. This is where the side-channel shines; with the extra information from it, we can now successfully predict all classes.

For the Complete FMNIST dataset, we add a few more concepts to the hierarchical tree to make it a complete set. These concepts represent seasonality and are: {Summer, Winter, Mild Seasons}. We consider them to be mutually exclusive. The updated dependency graph is shown in Fig. 1.

**Bird identification (CUB)**  The Caltech-UCSD Birds-200-2011 (CUB) [7] dataset Wah et al. (2011) consists of 11,788 natural images spanning 200 bird species. Each image is annotated with 312 binary concepts describing visual characteristics such as color, shape, length, pattern, and size. Following (Koh et al., 2020), we process concept labels via majority voting across instance-level annotations and remove excessively sparse concepts, resulting in $K = 112$ concepts. These binary concepts originate from categorical attributes, leading to 27 mutually exclusive groups, similar to one-hot encoded features. To define concept relationships in $G_C$, we assume that concepts related to the same type of attribute (e.g., all colors, all patterns) are interconnected via bidirected edges i.e. the concepts that pertain to the same "type" of feature are related, but we do not know its direction. These types of features are: {Shape, Color, Length, Pattern, Size}. Thus, the colors, patterns, etc. of all body parts are related. This structured representation embeds reasoning by enforcing relationships among semantically related features. Furthermore, we assume that all concepts directly influence all classes, ensuring that the model can fully leverage fine-grained feature representations.

**Smile Detection (CelebA)**  The CelebA [8] dataset (Liu et al., 2015) comprises over 200.000 celebrity face images annotated with 40 facial attributes. We select $K = 7$ facial attributes as concepts:

---

[6] https://github.com/zalandoresearch/fashion-mnist, MIT License

[7] https://www.vision.caltech.edu/datasets/cub_200_2011/

[8] https://mmlab.ie.cuhk.edu.hk/projects/CelebA.html

({Arched Eyebrows, Bags Under Eyes, Double Chin, Mouth Slightly Open, Narrow Eyes, High Cheekbones, Rosy Cheeks}), and use "Smiling" as the target label, making it a binary classification problem. The corresponding reasoning graph ($G$) is illustrated in Figure 3.

# D   Implementation Details

This section provides additional details on the implementation of our experiments. A more detailed version of CREAM's illustration is shown in Fig. 9. For datasets containing mutually exclusive concepts, we apply a softmax activation as described in Section 4. Our framework is implemented in PyTorch (v2.4.0) (Paszke et al., 2019) and PyTorch Lightning (v2.3.0). We use 20% of the stored activation values to perform the missing value imputation in CCI, and set SAGE's convergence threshold to $5 \times 10^{-2}$. The Permutation Feature Importance (PFI) metric is computed by permuting the channel values 100 times. All reported experiments use five different seeds. Lastly, for all ablation studies including the dropout rate, we use these $p$ values: {0.0001, 0.1, 0.2, 0.3, 0.4, 0.5, 0.6, 0.7, 0.8, 0.9, 0.99}.

**Details per Dataset**   Our experiments utilize three datasets, each with tailored architectures and preprocessing steps. We use the Adam optimizer (Kingma, 2014) for all models. For FashionMNIST (FMNIST), we employ a lightweight CNN backbone with two convolutional layers, ReLU activations, Max Pooling, dropout, and a final linear layer. The dataset is split into $50k - 5k - 10k$ for training, validation, and testing, respectively. The backbone is trained for 50 epochs, using a learning rate of $10^{-3}$ and a batch size of 256. Standard normalization is applied to the dataset. In the experiments of the main text, the number of epochs was also set to 50. Lastly, the standard model was trained from scratch for the same number of epochs.

For CUB and CelebA, we use ImageNet ResNet-18 (He et al., 2016) as the backbone. CUB follows the same train-test splits, concept processing, and image preprocessing as in (Koh et al., 2020). The backbone is fine-tuned for 50 epochs, with a learning rate of $10^{-4}$ and a batch size of 64. For CelebA, we fine-tune ImageNet ResNet-18 on a 5K image subset for 90 epochs, using a learning rate of $10^{-4}$ and a batch size of 256, following (Yang et al., 2022). Both datasets undergo identical preprocessing: color jittering, random resized cropping, horizontal flipping, and normalization. In the experiments of the main text, the number of epochs for CUB was 300, while in CelebA the maximum number of epochs was 200.

**Model Selection**   In all cases, the side-channel consists of a Linear layer with a ReLU activation. We perform grid search over hyperparameters (Table 6). The Masked MLP depth refers to the number of hidden layers in the masked algorithm from Zuko (Rozet et al., 2022); a depth of zero means the Masked MLP is equivalent to a Masked Linear layer. Given that CBMs require multiple selection criteria, we adopt a ranking-based approach that averages performance across task and concept accuracies on the validation set, as in (Sanchez-Martin et al., 2024). Table 7 reports the best hyperparameter configurations for each dataset. Note that we eventually selected the models with the highest dropout rate in each case. Here, $d_C K + d_Y$ represents the total latent space dimension, while $d_Y$ refers to the side-channel input size. The depth (number of hidden layers) of the masked MLP is $d$. For the CBM model, in both iFMNIST and cFMNIST we used a learning rate of $10^{-2}$. For the $C_{true} \to Y$ model we trained linear classifiers for all datasets. Specifically, for iFMNIST and CelebA we tried a MLP with ReLUs and in total 3 layers, to try to improve the upper bound in task accuracy.

Table 6: Hyperparameter search space explored for each dataset.

| Dataset | iFMNIST | cFMNIST | CUB | CelebA |
|---|---|---|---|---|
| Dropout ($p$) | $\{0.2, 0.5, 0.8, 0.9\}$ | $\{0.2, 0.5, 0.8\}$ | $\{0.1, 0.2, 0.5, 0.8\}$ | $\{0.2, 0.5, 0.8\}$ |
| $d_C K + d_Y$ | $\{8, 18, 76, 78, 128\}$ | $\{11, 21, 22, 32, 43, 128\}$ | $\{424, 512, 648, 848\}$ | $\{7, 8, 10, 36, 70, 75, 256, 512\}$ |
| $d_Y$ | $\{0, 10, 20, 30, 64\}$ | $\{0, 10, 40, 62\}$ | $\{64, 176, 200, 400\}$ | $\{0, 1, 3, 5, 123, 162\}$ |
| $\lambda$ | 1 | 1 | 1 | 0.25, 0.75, 1 |
| $d$ | $\{0, 2\}$ | $\{0, 2\}$ | $\{0, 3\}$ | $\{0, 5\}$ |

**Minimum Hardware Requirements**   All experiments were conducted on a high-performance computing cluster with automatic job scheduling, ensuring efficient resource allocation. We list the minimum hardware requirements we used. For FMNIST (both iFMNIST and cFMNIST), we utilized 2 CPU workers, 8GB RAM, and a GPU with at least 4GB of VRAM. For CUB, the setup included

Table 7: Configurations for the best-performing models in each dataset. Note that $d_C K + d_Y$ represents the dimensionality of latent space that is split, and $d_Y$ the dimensionality of the input to the side-channel. H-CREAM and S-CREAM refers to the models with hard and soft concept representations respectively. The best models have $d = 0$ number of hidden layers in the Concept-Concept Block.

| Dataset | Model | $\lambda$ | lr | $p$ | $d_C K + d_Y$ | $d_Y$ |
|---------|-------|-----------|------|-----|---------------|-------|
| iFMNIST | H-CREAM | 1 | 1e-3 | 0.9 | 78 | 30 |
|         | S-CREAM | 1 | 1e-3 | 0.9 | 76 | 20 |
| cFMNIST | H-CREAM | 1 | 1e-3 | 0.8 | 43 | 10 |
|         | S-CREAM | 1 | 1e-3 | 0.8 | 128 | 40 |
| CUB | S-CREAM | 1 | 1e-4 | 0.8 | 648 | 200 |
| CelebA | S-CREAM | 1 | 1e-3 | 0.1 | 75 | 5 |

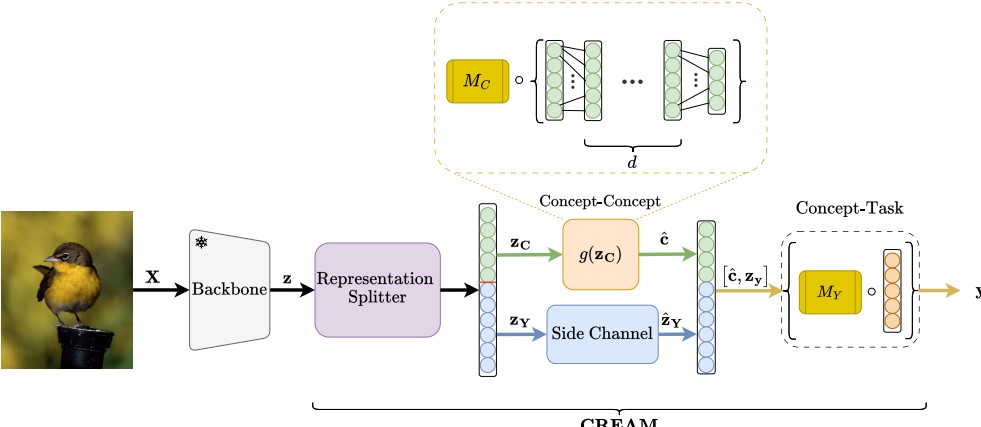

Figure 9: Expanded version of the illustration of CREAM found in the main text, providing additional details for clarity and completeness.

8 CPU workers, 16GB RAM, and a GPU with at least 4GB of VRAM. The CelebA experiments were more resource-intensive, requiring 12 CPU workers, 32GB RAM, and a GPU with at least 8GB of VRAM. The aforementioned resources refer to CBM and CREAM. A computational efficiency comparison can be found in App. E.1.

# E    ADDITIONAL RESULTS

## E.1    EFFICIENCY

All experiments were conducted on a high-performance computing cluster with automatic job scheduling, ensuring efficient resource allocation. For the efficiency results we set specific hardware to ensure fair comparisons. For models using a GPU we set the hardware requirements to: 96GBs RAM, 8 cores out of an AMD EPYC 7662 64-Core Processor CPU, NVIDIA A100-PCIE-40GB GPU. For the CPU entries we set the hardware requirements to: 256GBs of RAM and we used all cores of a AMD EPYC 9654 96-Core Processor CPU. To ensure a correct comparison by avoiding cache speedup, we used 5 burn-in iterations, and then for each model we report the mean values across 20 iterations. We also used 1 dataloader worker. We ensured correct timings and memory were recorded via using CUDA events, and tracemalloc in the CPU case. We also manually controlled the garbage collector.

We observe that across all datasets and devices, CREAM is relatively more computationally efficient than the rest of the models. Also, we notice that adding the side-channel to CBM (CBM+SC) only slightly decreases efficiency, supporting our claims about its efficiency.

Table 8: Comparison of Training Time and GPU Peak Memory Usage relative to CBM. Values indicate the factor difference for models (e.g., CREAM takes 1.8× longer to train than the baseline).

| Model | iFMNIST | | cFMNIST | | CUB | | CelebA | |
|---|---|---|---|---|---|---|---|---|
| | Time | Peak Memory | Time | Peak Memory | Time | Peak Memory | Time | Peak Memory |
| **CBM** | $x1.000_{0.000}$ | $x1.000_{0.000}$ | $x1.000_{0.000}$ | $x1.000_{0.000}$ | $x1.000_{0.000}$ | $x1.000_{0.000}$ | $x1.000_{0.000}$ | $x1.000_{0.000}$ |
| **CBM+SC** | $x1.496_{0.137}$ | $x1.001_{0.000}$ | $x1.631_{0.208}$ | $x1.001_{0.000}$ | $x1.017_{0.026}$ | $x1.004_{0.000}$ | $x1.004_{0.003}$ | $x1.001_{0.000}$ |
| **ACBM** | $x5.304_{0.535}$ | $x1.044_{0.000}$ | $x7.896_{0.900}$ | $x1.044_{0.000}$ | $x14.125_{0.140}$ | $x1.248_{0.000}$ | $x3.269_{0.050}$ | $x1.072_{0.000}$ |
| **SCBM** | $x27.535_{2.817}$ | $x1.050_{0.000}$ | $x27.985_{3.721}$ | $x1.053_{0.000}$ | $x9.210_{0.258}$ | $x1.303_{0.000}$ | $x3.269_{0.050}$ | $x1.072_{0.000}$ |
| **C$^2$BM** | $x12.194_{0.182}$ | $x1.003_{0.000}$ | $x15.143_{1.430}$ | $x1.005_{0.000}$ | - | - | $x2.846_{0.092}$ | $x1.938_{0.000}$ |
| **CREAM** | $x1.816_{0.159}$ | $x1.000_{0.000}$ | $x1.940_{0.048}$ | $x1.001_{0.000}$ | $x2.461_{0.212}$ | $x1.004_{0.000}$ | $x1.004_{0.006}$ | $x1.001_{0.000}$ |

Table 9: Comparison of Training Time (CPU) and overall system Peak Memory Usage relative to CBM. Values indicate the factor difference for models (e.g., CREAM takes 1.8× longer to train than the baseline).

| Model | iFMNIST | | cFMNIST | | CelebA | |
|---|---|---|---|---|---|---|
| | Time | Peak Memory | Time | Peak Memory | Time | Peak Memory |
| **CBM** | $x1.000_{0.000}$ | $x1.000_{0.000}$ | $x1.000_{0.000}$ | $x1.000_{0.000}$ | $x1.000_{0.000}$ | $x1.000_{0.000}$ |
| **CGM$_{CD}$** | $x9.150_{1.033}$ | $x361.874_{2.061}$ | $x20.238_{3.084}$ | $x1139.224_{3.005}$ | $x2.209_{0.070}$ | $x82.571_{2.607}$ |
| **CGM$_{prior}$** | $x8.471_{3.195}$ | $x36.138_{0.405}$ | $x8.465_{0.969}$ | $x40.627_{0.351}$ | $x2.227_{0.035}$ | $x18.635_{1.404}$ |
| **CREAM** | $x5.387_{1.437}$ | $x1.259_{0.009}$ | $x6.609_{2.438}$ | $x1.259_{0.042}$ | $x1.963_{0.021}$ | $x1.122_{0.003}$ |

### E.2 PERFORMANCE WITH AND WITHOUT SIDE-CHANNEL

In this section we showcase the modularity of the side channel. Specifically, we remove it from CREAM and add it a CBM. From the hyperparameter search performed in Section D, we pick the best models without a side channel ($d_Y = 0$ values from Table 6) for CREAM. The hyperparameters of these models are seen in Table 10.

Table 10: Hyperparameter configurations for the best-performing models (without a side-channel) in each dataset. The best models have $d = 0$ number of hidden layers in the Concept-Concept Block.

| Dataset | Model | $\lambda$ | lr | $d_C K + d_Y$ |
|---|---|---|---|---|
| iFMNIST | CREAM | 1 | 1e-3 | 128 |
| cFMNIST | CREAM | 1 | 1e-3 | 22 |
| CUB | CREAM | 1 | 1e-4 | 112 |
| CelebA | CREAM | 0.75 | 1e-3 | 7 |

We report the concept and task accuracies of CBM and CBM with a side-channel (CBM+SC), and CREAM without a side-channel and with the side-channel (CREAM+SC). The results are reported in Table 11. According to Table 11, CREAM outperforms both CBMs. Also we notice that, including the side-channel almost always leads to increased task and concept accuracy, in both CBM and CREAM. The former verifies our claims about the side-channel, while the latter is an added bonus. We believe it originates from ease of optimization.

Table 11: Task and concept accuracy(%). Reported values represent the mean and standard deviation. CREAM consistently achieves a better balance between performance and interpretability. The best-performing method is highlighted in **bold** and the second-best is underlined.

| Model | iFMNIST | | cFMNIST | | CUB | | CelebA | |
|---|---|---|---|---|---|---|---|---|
| | $ACC_Y$ | $ACC_C$ | $ACC_Y$ | $ACC_C$ | $ACC_Y$ | $ACC_C$ | $ACC_Y$ | $ACC_C$ |
| **Black-box** | $92.70_{0.07}$ | - | $92.70_{0.07}$ | - | $74.85_{0.12}$ | - | $78.47_{0.30}$ | - |
| $C_{true} \rightarrow Y$ | 60.00 | - | 100.00 | - | 100.00 | - | 84.51 | - |
| **CBM** | $\underline{91.19}_{0.40}$ | $96.74_{0.36}$ | $92.00_{0.17}$ | $97.33_{0.06}$ | $\underline{73.76}_{0.32}$ | $80.90_{0.08}$ | $79.28_{0.39}$ | $79.77_{0.17}$ |
| **CBM+SC** | $91.02_{0.20}$ | $96.13_{0.20}$ | $92.18_{0.11}$ | $97.38_{0.08}$ | $\mathbf{74.36}_{0.10}$ | $82.79_{0.11}$ | $79.55_{0.24}$ | $\mathbf{79.97}_{0.21}$ |
| **CREAM** | $57.10_{0.05}$ | $\mathbf{99.07}_{0.02}$ | $\underline{92.31}_{0.15}$ | $\underline{97.52}_{0.25}$ | $71.13_{0.22}$ | $\underline{85.66}_{0.09}$ | $\underline{80.69}_{1.13}$ | $78.46_{1.50}$ |
| **CREAM+ SC** | $\mathbf{92.43}_{0.23}$ | $\mathbf{99.07}_{0.03}$ | $\mathbf{92.38}_{0.16}$ | $\mathbf{98.08}_{0.06}$ | $72.90_{0.28}$ | $\mathbf{86.83}_{0.04}$ | $\mathbf{80.92}_{0.55}$ | $\underline{79.91}_{0.22}$ |

We also study their intervenability. The results, visualized in Fig. 10 are similar to the ones in the main paper. However, we also notice that including a side-channel seems to alleviate the drop in accuracy by intervening in CUB.

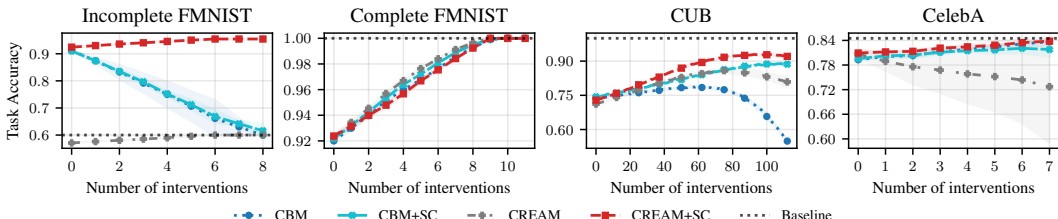

Figure 10: Impact of individual interventions on task accuracy. We reach similar conclusions as the ones in the main text.

### E.3 GROUP INTERVENTIONS

In this section we examine the benefits of the C-C mutually exclusive relationships, on the efficiency of interventions. Specifically, we investigate group interventions, where contrary to individual interventions, the human expert intervenes on a group of mutually exclusive concepts simultaneously. For instance, in cFMNIST, a human expert would change the value of all concepts belonging in Mutex 3 ("Summer", "Winter" and "Mild Seasons") with just one intervention, by "activating" the concept "Summer" and the rest being deactivated automatically, since they are mutually exclusive. Note that, in the case of group interventions the upper bound of interventions drops even lower, down to the number of directly connected concept mutex groups. As visualized in Fig. 11, in FMNIST, we observe that task performance peaks after one group intervention in the incomplete setting and two group interventions in the complete setting. For both cases, the one remaining intervention, does not improve accuracy since those are the indirect mutex concepts ("Clothes" and "Goods"). Similar conclusions are seen in CUB; with about 20 group interventions we achieve performance gains of almost 90 individual interventions. Lastly, group interventions lead to the same performance gains as individual interventions, but with less human effort. These findings indicate that group interventions on mutually exclusive concepts, identified through C-C relationships, can help scale the intervention procedure.

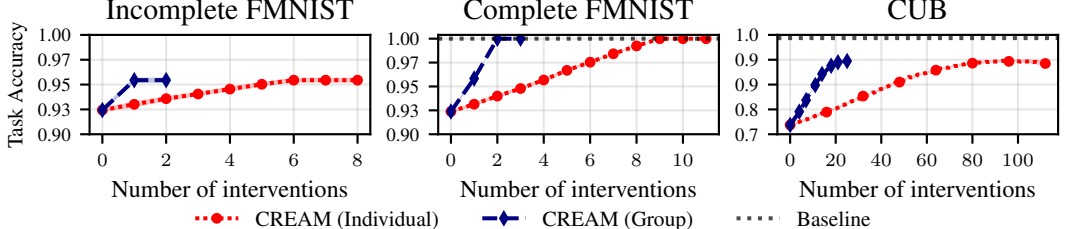

Figure 11: Impact of group interventions compared to individual ones, on task accuracy. Group interventions lead to the same performance gains with lower number of interventions. The number of relevant group interventions in CREAM for each dataset are, from left to right, 1, 2 and 27. Note the baseline's performance is not visualized in iFMNIST, as it is comparatively too low.

### E.4 CORRELATION OF VARIABLES

In an effort to investigate the effect of different types of relationships on the model representations, we investigate their statistical relationships. Specifically, we visualize the correlations between the concept exogenous variables and the input to the side-channel. As seen in Fig. 12, in the FMNIST cases, where the C-C relationships create a DAG, the exogenous variables of the concepts form blocks in the correlation matrix. Each of the $d_C$ dimensions seem to be strongly correlated with the rest of the dimensions of the exogenous variable they belong to. Furthermore, we can observe the hierarchical structure of the concepts. For example, the exogenous of "Goods" is also strongly correlated to "Accessories" and "Shoes". Meanwhile, in CUB, where the C-C relationships create cycles in $G_C$, we do not observe any particular structure. In addition, CelebA exhibits a lot of variables with zero variance; they represent "dead" neurons. Lastly, in all cases, the side-channel also does not show a particular structure.

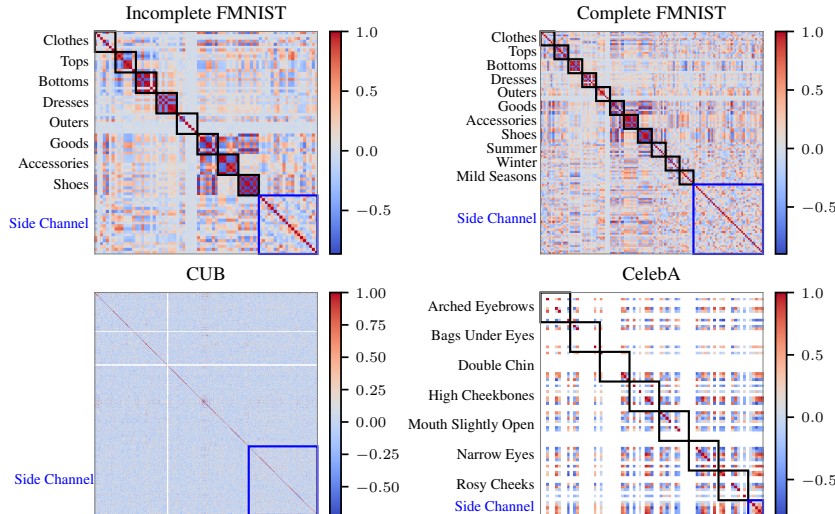

Figure 12: Correlation matrix for the CREAM models reported in Table 3. For each concept we draw a box around its assigned $d_C$ dimensions. Note that CUB's exogenous are too small to plot. We notice that `C-C` reasoning leads block-like correlations, both within the exogenous of each concepts, and between their exogenous, revealing the hierarchical structure.

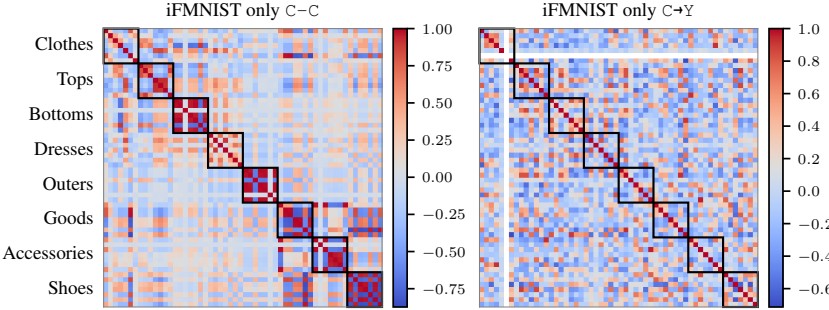

Figure 13: Correlation matrix for the CREAM models reported in Table 3. For each concept we draw a box around its assigned $d_C$ dimensions. We notice that `C-C` reasoning leads block-like correlations, both within the exogenous of each concepts, and between their exogenous, revealing the hierarchical structure.

**Further investigating the block-diagonal structure**  We will now focus on identifying the cause of the near block-diagonal structure in the correlation matrices. Following the leakage experiments from Table 3, we visualize the correlations when the model reasoning is removed. According to Fig. 13, we identify `C-C` as the architectural choice leading to the near block-diagonal structure of the correlation matrix.

### E.4.1 VARIANCE OF THE SIDE CHANNEL

Here, we will also check if some side-channel nodes have turned into constants, since we set the output of the side-channel to be of dimension $L$, which leads to an excess of needed nodes. For



Figure 14: Covariance matrix of the outputs of the side-channel for CREAM. On the left, we report the corresponding class each node is assigned to. CUB has too many classes to clearly visualize.

instance, in the iFMNIST dataset, the classes: {Trouser, Dress, Coat, Bag} are fully distinguishable from their assigned concepts, meaning that they do not need the side-channel. Thus, it would be preferable if their assigned side-channel nodes had a constant output. To verify this, we visualize the covariance matrix of the side-channel ($\hat{\mathbf{z}}_Y$), and we inspect its main diagonal. Specifically, for CelebA, $L = 1$ and thus there is only one side-channel node, we report here its variance: $Var_{\hat{\mathbf{z}}} = 0.0$, meaning that its a constant value. As seen in Fig. 14, some of the nodes in iFMNIST (e.g., for "Trouser") and cFMNIST (e.g., for "Bag") have a low variance, meanwhile the classes that cannot be predicted only via concepts (e.g.,"T-shirt", "Shirt" , and "Ankle boot") have high variance, suggesting that CREAM primarily uses the side-channel information to predict only these classes.

### E.5 CAUSAL DISCOVERY AND CREAM

One key distinction between CREAM and CGM Dominici et al. (2025) lies in how the reasoning graph is obtained. As noted previously, $CGM_{CD}$ employs causal discovery algorithms to infer the underlying causal (reasoning) structure from observational data. Thus, we will use one of these discovered graphs from $CGM_{CD}$ to simulate a causal discovery scenario. The discovered PDAG is visualized in Fig. 15. We will model, the undirected edges of the PDAG as symmetric entries in $A_C$, as we mentioned in Section 3.1. Using the same hyperparameters as the ones in the main text, we report: $ACC_Y = 79.56_{2.19}$, and $ACC_C = 79.55_{0.83}$. Thus, CREAM exhibits similar performance when using our DAG.

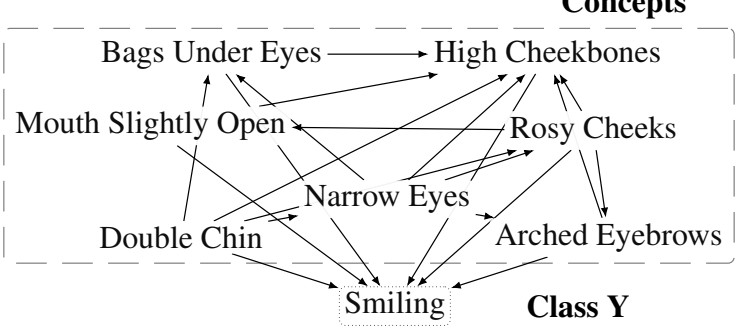

Figure 15: Reasoning graph taken from one the $CGM_{CD}$ runs Dominici et al. (2025) for predicting "Smiling" in CelebA (Liu et al., 2015).

## F ABLATION STUDIES

In this section, we present additional experiments to offer a deeper understanding of CREAM's hyperparameters. We conduct ablation studies on multiple hyperparameters to illustrate their impact on different aspects of model performance, supplementing the main text with further insights.

### F.1 EFFECT OF DIMENSIONALITY OF EXOGENOUS VARIABLES

As mentioned in Section 4.2 each exogenous variable $\mathbf{z}_C$ is of dimension $d_C$. Here we study the effect of $d_C$ on model performance (as seen in Fig. 16), for $d_C \in \{1, 2, 3, 7, 10\}$. Note that, we keep every other hyperparameter to the same values as in Table 7. This means, we are increasing the dimensionality of the output of the representation splitter, while keeping $|\mathbf{z}_Y|$ constant.

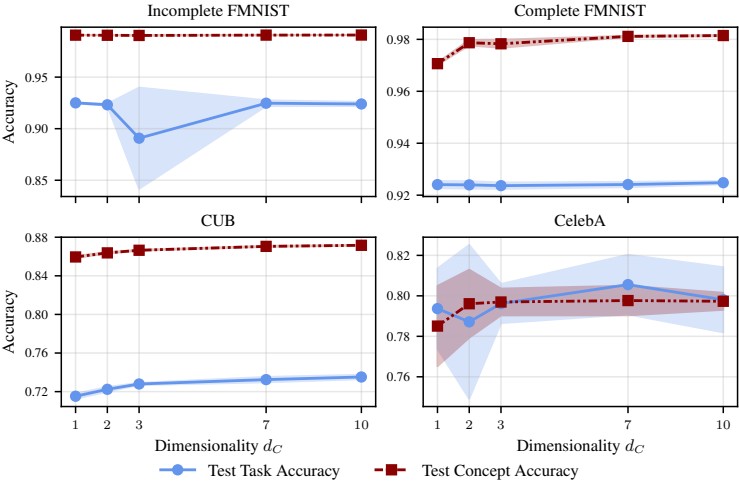

Figure 16: Concept and Task accuracies across different $d_C$ values for the exogenous of the concepts. Values are presented as mean $\pm$ standard deviation. Increasing $\mathbf{z}_Y$, generally improves performance.

We observe a general trend across all datasets; increasing the dimensionality $d_C$ of the exogenous variables of the concepts leads to same or slightly increased performance. That increase of performance is sometimes in the form of increased concept accuracy, task accuracy, or both at the same time.

### F.2 EFFECT OF THE DEPTH OF CONCEPT-CONCEPT BLOCK

In Fig. 17 we study the effect of increasing the depth $d$ of the Concept-Concept Block to model performance. For each model we keep the same configuration as the main text, but now $d \in \{0, 1, 2, 3, 5\}$. We notice a trend across all models; increasing the depth leads to equal or worse performance. These findings further support the notion that human-interpretable concepts are often linearly encoded in the latent space of neural networks (Rajendran et al., 2024).

## G HARD MODELS

In this section we will study CREAM with hard concept representations, i.e., $\hat{C} \in \{0, 1\}$. To ensure trainability in the hard case, we leverage the straight-through estimator (STE) (Bengio et al., 2013), which approximates gradients during the backward pass. The representation of hard and soft concepts is given as:

$$\hat{C} := \begin{cases} \sigma(\hat{l}_C) & \text{S-CREAM} \\ round(\sigma(\hat{l}_C)) & \text{H-CREAM} \end{cases} \tag{17}$$

where H-CREAM and S-CREAM are for hard and soft concepts respectively. We also handle mutually exclusive concepts similarly. The hyperparameter configurations of H-CREAM, can be found in Table 7. Hard CBM models are sometimes preferred due to them not suffering from leakage (Havasi et al., 2022; Vandenhirtz et al., 2024) and being easier to intervene on.

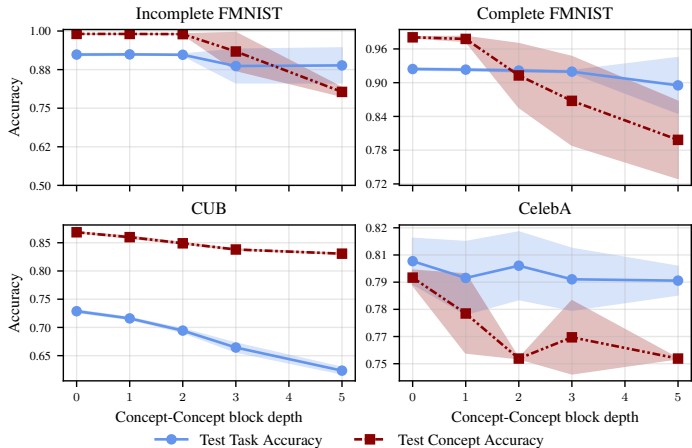

Figure 17: Concept and Task accuracies across different $d$ values in the Concept-Concept block. Values are presented as mean $\pm$ standard deviation. Increasing $d$ of the `C-C` block degrades performance.

## G.1 Concept and Task Accuracy

S-CREAM consistently outperforms its hard variant across all datasets and evaluation metrics. This suggests that soft concept representations provide greater flexibility and are easier optimize, allowing the model to capture useful variations in concept values while maintaining structured reasoning.

Table 12: Performance comparison between CREAM variants. Reported values represent the mean and standard deviation over five seeds. S-CREAM outperforms H-CREAM performance and interpretability in both FMNIST variants. The best-performing model variant is highlighted in **bold**.

| Dataset | Model | Test $ACC_Y$ | Test $ACC_C$ |
|---------|-------|--------------|--------------|
| iFMNIST | S-CREAM | $\mathbf{92.43}_{0.23}$ | $\mathbf{99.07}_{0.03}$ |
|         | H-CREAM | $92.15_{0.17}$ | $98.98_{0.01}$ |
| cFMNIST | S-CREAM | $\mathbf{92.38}_{0.16}$ | $\mathbf{98.08}_{0.06}$ |
|         | H-CREAM | $91.29_{0.73}$ | $96.70_{0.17}$ |

## G.2 Importances

Table 4 demonstrates that, CCI remains consistently above the critical $0.5$ threshold, and PCI exceeds PSI, for the selected hyperparameter settings. This indicates that the hard variant can also fulfill our concept importance desiderata. Furthermore, soft models report a higher CCI value compared to their hard counterparts, indicating that their decisions are more strongly influenced by the concepts.

## G.3 Interventions

We compare task accuracy between the two CREAM variants, after individual concept interventions. We follow the same process as the one mentioned in Section 5. For H-CREAM we do not have to find the 5th and 95th percentiles, and we instead directly use the ground truth concepts. As seen in Fig. 18, H-CREAM shows the same behavior as S-CREAM. However, the latter slightly outperforms the former in both presented cases.

## G.4 Correlations

We also investigate the correlations between the concept exogenous variables and the input to the side-channel in the hard variant. As illustrated in Fig. 19, H-CREAM does not exhibit the block correlations that the soft model does (Fig. 12). Also, in H-CREAM more concept exogenous variables

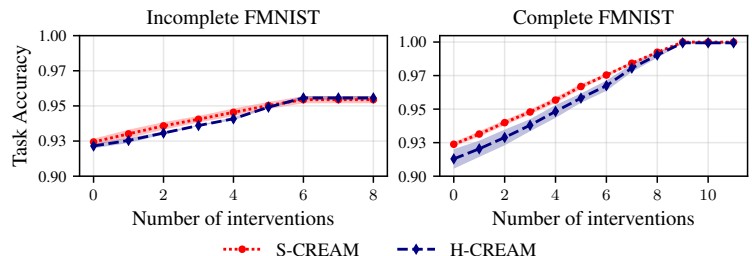

Figure 18: Intervenability comparison of H-CREAM and S-CREAM. The latter slightly outperforms the latter. Both models' task accuracy peaks after 6 and 9 interventions in iFMNIST and cFMNIST respectively.

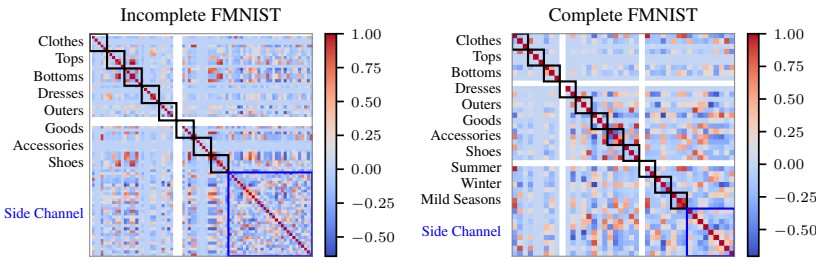

Figure 19: Correlation matrices of the output of the representation splitter ($\mathbf{z}$) for H-CREAM. We notice that more concept exogenous variables have lowered correlations compared to the soft models.

have become constants. This indicates that the hard variant requires a lower number of exogenous variables, which is also in line with the best hyperparameters seen in Table 7.

## H  LOGIC VIEWPOINT OF CREAM

In this section we will express the relationships in $G$ using description logic (Baader, 2003). Note that in CREAM the concepts are calculated using the exogenous variables, and the tasks using the concepts and their corresponding side-channel input. Table 13 shows some of the logic rules that match CREAM's calculations, with a slight abuse of notation. For instance, a mutex constraint is described as: Clothes $\sqcap$ Goods $\sqsubseteq \bot$, and concepts are calculated by: Tops $\leftarrow \mathbf{z}_{Clothes} \sqcap \mathbf{z}_{Tops}$.

## I  THE USE OF LARGE LANGUAGE MODELS (LLMS)

The authors used a large language model (LLM) as a general-purpose writing assistant during the preparation of this paper. Its application was exclusively for grammar, spelling, and word choice. This work did not involve use of LLMs for core methodology, scientific rigorousness, or originality of research.

Table 13: Some logic rules based on description logic for CREAM. The **z** variables correspond to the concept and side-channel, and $\perp$ denotes the empty set.

| Dataset | |
|---|---|
| iFMNIST | Clothes $\leftarrow \mathbf{z}_{Clothes}$ 
 Goods $\leftarrow \mathbf{z}_{Goods}$ 
 Clothes $\sqcap$ Goods $\sqsubseteq \perp$ 
 Tops $\sqcap$ Bottoms $\sqsubseteq \perp$ 
 Tops $\sqcap$ Dresses $\sqsubseteq \perp$ 
 $\vdots$ 
 Accessories $\sqcap$ Shoes $\sqsubseteq \perp$ 
 Tops $\leftarrow \mathbf{z}_{Clothes} \sqcap \mathbf{z}_{Tops}$ 
 $\vdots$ 
 Shoes $\leftarrow \mathbf{z}_{Goods} \sqcap \mathbf{z}_{Shoes}$ 
 T-shirt $\sqcap$ Pullover $\sqsubseteq \perp$ 
 T-shirt $\sqcap$ Shirt $\sqsubseteq \perp$ 
 $\vdots$ 
 Sneaker $\sqcap$ Ankle Boot $\sqsubseteq \perp$ 
 T-shirt $\leftarrow$ Tops $\sqcap \mathbf{z}_{T-shirt}$ 
 $\vdots$ 
 Ankle Boot $\leftarrow$ Shoes $\sqcap \mathbf{z}_{AnkleBoot}$ |
| CelebA | Bags Under Eyes (BUE) $\leftarrow \mathbf{z}_{BUE}$ 
 High Cheekbones (HC) $\leftarrow \mathbf{z}_{HC}$ 
 Mouth Slightly Open (MSO) $\leftarrow \mathbf{z}_{MSO}$ 
 Rosy Cheeks (RC) $\leftarrow \mathbf{z}_{RC}$ 
 Arched Eyebrows (AE) $\leftarrow \mathbf{z}_{AE}$ 
 Double Chin (DC) $\leftarrow \mathbf{z}_{DC}$ 
 Narrow Eyes (NE) $\leftarrow \mathbf{z}_{MSO} \sqcap \mathbf{z}_{BUE} \sqcap \mathbf{z}_{HC} \sqcap \mathbf{z}_{NE}$ 
 Smiling $\leftarrow BUE \sqcap HC \sqcap MSO \sqcap NE \sqcap RC \sqcap DC \sqcap AE \sqcap z_{Smiling}$ |

