# OpenReview forum: "Towards Reasonable Concept Bottlenecks"
_ICLR.cc/2026/Conference — ICLR 2026 Conference Withdrawn Submission_

### Official Review · Reviewer_PXjB · 2025-10-25

**Soundness:** 3
**Presentation:** 3
**Contribution:** 1
**Rating:** 4
**Confidence:** 4

**Summary:**

This paper introduces Concept REAsoning Models (CREAMs). CREAMs are hierarchical concept-based reasoning models that can incorporate human knowledge about their task in the form of concept-to-concept and concept-to-task relationships. CREAM can then adapt to concept-incomplete tasks by introducing a regularized information side channel without sacrificing intervention or task performance. This work evaluates CREAM against a series of hierarchical baselines (e.g., CGMs and $\text{C}^2$BMs) across four tasks, demonstrating that CREAM achieves promising task, concept, and intervention performance.

**Strengths:**

Thank you so much for submitting this work! I enjoyed reading this paper, learned a lot from it, and appreciate the time taken to write it up and submit it to ICLR. Below are what I believe are this paper’s main strengths:

1. **[Quality, Critical]** The proposed methodology is technically sound and clearly motivated. Moreover, the authors provide a substantial appendix with ablations, runtime analyses, and qualitative insights that pre-empt many typical questions one may have about hyperparameters and design choices.
2. **[Significance, Major]** The problem addressed, namely that of explicitly integrating human knowledge about concept dependencies into CBMs, is timely and relevant to several communities. This includes the XAI, causal reasoning, graph learning, and neurosymbolic communities. As such, I believe this work has the potential to interest a large contingent of ICLR.
3. **[Originality, Minor]** CREAM is certainly novel in its way of introducing, say, a StrNN-based approach for enforcing relationships to be respected and its use of hierarchical mutex relationships. However, its novelty can be considered somewhat limited, given similar prior works, such as CGM $\text{C}^2$BM, which highly overlapping aims and approaches. Because of this, I am marking its originality "minor" (see below for further details).
4. **[Clarity, Major]** The paper is very well-written and easy to follow. Specifically, all the main components are clearly motivated, the figures and tables are straightforward to read, and the text flows very naturally.

**Weaknesses:**

In contrast, I believe the following are some of this work’s limitations:

1. **[Quality, Critical]** The evaluation against CGMs and $\text{C}^2$BMs has some potential issues, which *may* make the comparisons in the paper not entirely fair. Please see the questions below for specific issues.
2. **[Significance, Critical]** The need for human knowledge when constructing CREAM is a severe limitation to the applicability of this work in real-world tasks where, say, concepts are discovered and relationships may be unknown (e.g., in the label-free style [1]). Therefore, it would be very helpful to understand how CREAM could be used in this setup. Moreover, the lack of inclusion of a real-world incomplete dataset, even though incompleteness is a key motivation for part of CREAM, limits the significance of the observed results. See below for specific questions on this matter.
3. **[Originality, Major]** Practically speaking, CGMs and $\text{C}^2$BMs, which were used in this paper’s evaluation, can support similar structures as those used in CREAM, and they have very similar methodological approaches to the same task. Now they both achieve this in different ways (e.g., they do not use any StrNNs or regularized bypasses), so the methods are indeed different. However, the approaches are still very similar in nature, objectives, and the task they are trying to solve. In my opinion, the highly overlapping aims of all three of these methods make the novelty of this paper very limited. On this note, although the paper phrases $\text{C}^2$BM as "concurrent work", I would ask the authors to consider that this paper has been out for several months now (the first arXiv submission shows March 2025). Therefore, it is not within the "traditional" concurrence window of 3 months before submission.


### References
1. Oikarinen et al. "Label-free Concept Bottleneck Models." ICLR (2023).

**Questions:**

Currently, given my concerns with this work's novelty and evaluation, and considering them alongside the strengths I listed above, I am *softly* leaning towards rejection (score of 4). However, I am absolutely happy to be convinced that some or all of my conclusions are wrong and to change my recommendation based on a discussion with the authors. For this, the following questions could help clarify/question some of my concerns (if space/time is constrained, please focus on the critical questions):

1. **[Critical]** Could you please clarify why it is the case that you could not use GPUs for CGMs?  If I recall that work correctly, that is not a particular limitation of that work. In that case, I believe these methods should be trained on GPUs to that they can scale properly to the tasks where they are currently missing in the evaluation (e.g., CUB). Moreover, are the runtime values then compared between CPU-based models (e.g., CGMs) and GPU-based models (e.g., CREAM)? If so, is this a fair evaluation? This should be made explicitly clear if it is not the case.
2. **[Critical]** Related to the question above, why are $\text{C}^2$BMs incompatible for CUB? Notice that their own paper uses CUB as an evaluation task (granted, using a different concept relationship set). Still, it seems odd not to have either of the closest competitors (CGM and $\text{C}^2$BM) in one of the two real-world tasks used in evaluation.
3. **[Critical]** Something seems off with the CBMs in the experiments. For example, their interventions on CUB are significantly different to those observed in a lot of previous works (e.g., [1-3]). This makes me believe that something may be off in the evaluation setups. My hunch is that you are using a logit-based bottleneck?  Could you please clarify why this is the case and why this is the right evaluation setup if there is an intentional reason behind that result? Would it be possible to include in the main paper the results of the standard sigmoidal bottleneck CBM? Just a note that the sigmoidal CBM is what most of the literature usually refers to as a soft CBM (rather than the logit-based bottleneck).
4. **[Critical]** Similarly, I believe this work would benefit from evaluating on a more complex incomplete dataset (notice how the vanilla CBM performs very well on iFMNIST, indicating this is not a particularly difficult incomplete dataset). Do you have a sense of how CREAM performs on stronger benchmarks for incompleteness, such as, say, a version of CUB where several concept groups are not provided (e.g., as seen in [2-6])? Given the emphasis placed on incompleteness in this work, I believe having a good benchmark for this sort of task is important.
5. **[Major]** Could you explain how, if at all, CREAM can be used in setups where concept relationships may be unknown (e.g., label-free instances)? Would assuming a bipartite structure (a-la CBMs) would be something that could be used in those setups? If so, how does this model perform then? Notice that previous similar approaches, such as CGMs and $\text{C}^2$BMs, offer mechanisms for discovering underlying graph structures in an automated way.
6. **[Major]** I am not entirely convinced that CCI “values near 1 indicate more substantial reliance on concepts,  and thus higher interpretability.” I agree with the first part of that statement, but why wouldn't depending on a concept be **necessarily** less interpretable? If many concepts are not helpful for downstream task prediction, then the model not depending on them to make its prediction is not necessarily an indication of a lack of interpretability (since it is doing what it should, given the set of training concepts it had at its disposal). Could you please further elaborate on this?
7. **[Major]** In line 413, the paper mentions that “CGM again surpasses the $C_\text{true} \rightarrow Y$ performance, at the cost of interpretability”. Why does this cost interpretability if CGM’s underlying graph is as informative as that used in CREAM? More importantly, how does that “lack of interpretability” manifest in real-world consequences, which would be unique to CGM vs CREAM?
8. **[Minor]** In lines 200-202, the paper claims that “the number of effective interventions is lowered from $K$ to $|C_\text{direct}|$”. I think I get the intent of this statement (i.e., intervening on the immediate parent of a task and on the parent of that concept would nullify the intervention on the non-direct concept). However, wouldn’t one be able to propagate interventions across concepts in $C_\text{direct}$ by intervening on the non-direct concepts (in a similar fashion to ground-truth interventions on Causal CGMs)? If so, this is a strength rather than a limitation, and I find this original statement on the paper particularly odd/misleading, as it appears to imply one should only ever intervene on the concepts in $C_\text{direct}$.
8. **[Minor]** Related to the comment above, in lines 204-205, the paper says “$G_C$ highlights relationships among concepts, such as mutually exclusive groups,  facilitating grouped interventions rather than individual ones”. Wouldn’t one have to know the exclusive groups in the first place to build $G_C$? If so, I am not entirely convinced of the strength of this argument given that both this approach and previous approaches that perform group-level interventions assume that you know those groups.

### Other Suggestions and Typos

Whilst reading this work, I found the following potential minor issues/typos which may be helpful when preparing a new version of this manuscript:

1. **[Missing Connection, Major]** When discussing the effect of leakage in embedding-based representations and how it can be avoided (Section 2), it may be good to include recent work discussing how leakage may be controlled in embedding-based models [2]. Notice that the regularization strategy suggested there for the side-channel information is very similar to the dropout regularization used in this work. As such, it would be helpful to make these connections/differences explicit in the text.
2. **[Citation Style, Nit]** I would recommend merging the references in “(Samek et al., 2019) (Doshi-Velez & Kim, 2017; Lipton, 2018)” into a single reference block.
3. **[Clarity, Minor]** The sentence “In this work, we propose reasonable concept bottleneck models that are guided, but not strictly limited,  by the designer-picked or automatically discovered  C-C and C→Y relationships.” makes sense. Still, it took me a few tries to parse it because of its structure. I would suggest splitting it into two sentences or rewriting for clarity.

### References

1. Koh et al. "Concept bottleneck models." International conference on machine learning. PMLR, 2020.
2. Espinosa Zarlenga et al. "Learning to receive help: Intervention-aware concept embedding models." NeurIPS (2023).
3. Espinosa Zarlenga et al. "Avoiding Leakage Poisoning: Concept Interventions Under Distribution Shifts." ICML (2025).
4. He et al. "Chat-CBM: Towards Interactive Concept Bottleneck Models with Frozen Large Language Models." arXiv preprint arXiv:2509.17522 (2025).
5. Ragkousis et al. "Tree-based leakage inspection and control in concept bottleneck models." arXiv preprint arXiv:2410.06352 (2024).
6. Zabounidis  et al. "Benchmarking and enhancing disentanglement in concept-residual models." arXiv preprint arXiv:2312.00192 (2023).

---

### Official Review · Reviewer_W29B · 2025-10-27

**Soundness:** 1
**Presentation:** 3
**Contribution:** 2
**Rating:** 4
**Confidence:** 3

**Summary:**

The core goal of this work is to enable a domain expert to explicitly encode additional prior knowledge into a CBM through the use of a concept and task graph to encode constraints (sparse C->Y relationships, mutually exclusive C-C, hierarchical/correlative C-C relationships), and propose a novel architecture CREAM to enforce these constraints. They also utilize a regularized side-channel to improve performance on incomplete concept sets.

The authors highlight two main benefits of their approach: their model can avoid concept leakage allowing for better interventions, and achieve task performance on par with black-box models. They evaluate these claims empirically, comparing to vanilla CBMs as well as more recent hard/soft CBMs (ACBM, SCSBM, CGM, C2BM).

**Strengths:**

- The ability to encode additional prior knowledge into a CBM is very interesting, going beyond a simple sparse linear layer.
- Mitigating concept leakage is an exciting direction for ensuring CBMs mean what they say in safety critical domains.

**Weaknesses:**

The claims and emphasis of the paper don’t seem fully supported by the results.

- Performance of the model achieves nearly equal task performance (within 1-2%) to a vanilla CBM. Also the CBM results in the main table are reported without a side channel though CREAM is reported with a side channel. Task performance does not seem to benefit from the authors' contributions.
- The authors also show an intervention result, where CBM fails on iFMNIST/CUB with too many interventions. However, CREAM does not outperform the other baselines, and other work that focuses directly on interventions is not compared against [1].

My understanding of the main novelty of the paper is the utilization of the concept-concept block and concept-task block to enforce specific behavior based on domain knowledge. With that in mind, the benefits of structured reasoning seems underexplored in the main text. A few examples of questions that I believe would be beneficial to showing that this architectural change is useful.
- What concrete benefits does enforcing different C-C or C-Y relationships have on the model's downstream performance?
- What C-C or C-Y relationships are evaluated in the paper?
- Why does knowing that colors/shapes/etc. are correlated help the model make better decisions? Why can we not handle this correlations implicitly based on the training data?

[1] Espinosa Zarlenga, Mateo, et al. "Learning to receive help: Intervention-aware concept embedding models." Advances in Neural Information Processing Systems 36 (2023): 37849-37875.

**Questions:**

Some questions roughly in order of importance.

1. Table 3 is very interesting. I am confused what the removing C->Y reasoning baseline does in practice. From my understanding it seems the C->Y reasoning encodes sparse relationships, how does this mechanism work and why does it lead to zero leakage? Does removing C->Y reasoning offer any guarantees with respect to leakage, and would this result hold for different notions of leakage (OIS, NIS from Parisini et al., 2025)?
    - If other metrics are being utilized, it would also be interesting to see if something similar occurs on the other datasets.
2. I am also confused about the side-channel and dropout, as the utilization of a side-channel is not new. The authors utilize CCI to measure global feature importance and use this to select the dropout rate, but is dropout being applied to the side channel only? Is the claim that the authors' side-channel regularization technique better than other methods (ex. residual fitting as in [1])?
3. Does each direct concept condition on itself as well? How does the concept-concept network handle multi-level hierarchies? From my understanding in practice this amounts to each direct concept having a set of all its ancestors that it can condition on. Perhaps an example of the claim in line 202 would be helpful for seeing how this would work in practice?
    - The authors also say "In CUB and CelebA, all concepts are direct." Does the concept-concept block play no role on these datasets?
4. Generally I am confused about what these graphs look like in practice for the different datasets.

[1] Yuksekgonul, Mert, Maggie Wang, and James Zou. "Post-hoc concept bottleneck models." arXiv preprint arXiv:2205.15480 (2022).

---

### Official Review · Reviewer_13LE · 2025-10-30

**Soundness:** 3
**Presentation:** 4
**Contribution:** 2
**Rating:** 4
**Confidence:** 3

**Summary:**

This paper introduces Concept REAsoning Models (CREAMs), a novel Concept Bottleneck Model framework designed to enhance interpretable reasoning by architecturally encoding prior knowledge of concept-concept (C-C) and concept-task (C-Y) relationships. To maintain high accuracy, the model also incorporates a regularized side-channel intended to complement potentially incomplete concept sets, encouraging predictions to remain concept-grounded. Experiments show the resulting models support efficient and accurate interventions by avoiding leakage while achieving task performance comparable to black-box models.

**Strengths:**

- Related work is clearly discussed.
- The paper is well-written and all the techniques well motivated and discussed.
- The paper combines nice ideas from different approaches, like CBMs, SCMs, StrNN obtaining a novel method with comparing performances over classic problems of CBMs with related work.

**Weaknesses:**

- Even if only a sub-portion of the concepts has a direct effect on the task, I don't understand why it should be excluded to have interventions on C_undirect. Ineed, as often happen in reality it can be cheaper to provide an intervention on low-level concepts (in the sense of parents in the causal graph). Also intervening on a parent concept will create an impact on both other concepts and, in turn, this will affect also the final task. The goal of CBM is generally both to maximize concept and task accuracy, and intervening on C_undirect, at the same time you'll get both a correction factor over multiple children concepts and, e.g., children-of-children tasks.
- "Our results in Table 2 show that CREAM (...) outperforming concept-based baselines and black-box models." Actually only with CelebA, CREAM outperforms the black-box but there the best results are obtained by CGM. Honestly it's also a bit difficult this evaluation as the task on CelebA is not standard. So I'd rephrase the sentence to loyally comment the findings.
- "Interpretability in the presence of a side-channel." I don't get why the authors report these results only in the appendix given that they claim (rightly) that this is a crucial point for interpretability (that in turn is the crucial point to use CBMs). And what does it mean that "We show that CCI > 0.5 is enough for our desiderata to hold." Given that in the appendix it is claimed that "When CCI ≈ 0.5, it suggests that both the concept and side-channels contribute equally to the predictions."
- Even if the approach is interesting I found the results not properly convincing with respect to competitors.


Minor comments:
- "CBMs require supervision solely on the concepts while NeSy require knowledge in the form of logical programs. " This sentence seems wrong or a bit too simplistic, indeed CBMs generally requires both concepts and tasks supervisions, and there are many forms of NeSy AI (and strictly connected methods from starAI) approaches, often combining logic, probability and neural models. So requiring as additional knowledge a causal graph I think perfectly aligns with statistical relational models within NeSy.
-- The technical part is a bit hard to follow without having some preliminary knowledge about StrNN (however the relevant info are in the appendix).

**Questions:**

1) It's not clear to me the effect of the side-channel w.r.t. interventions. I mean, the information carried out by the side-channel can actually have a similar to the nuisance related to concept-leakage. Indeed, the Concept-Task block uses both the side-channel and the
concepts to predict the task.
2) Have the authors tried CREAM without the side channel systematically? How are the results? Why don't avoid using it given the fact this can be a limitation for interpretability. E.g. I understand table 3 does it and show that the results are worst than the competitors reported in table 2.

---

### Note · Authors · 2025-11-17

**Comment:**

Thank you to the reviewers for taking the time to evaluate our work. We appreciate your feedback, and will be withdrawing our submission.

**Withdrawal Confirmation:**

I have read and agree with the venue's withdrawal policy on behalf of myself and my co-authors.